# Phase diagram of early training dynamics in deep networks: effect of the learning rate, depth, and width

**Dayal Singh Kalra** [*†]
dayal@umd.edu

**Maissam Barkeshli** [*‡§]
maissam@umd.edu

## Abstract

We systematically analyze optimization dynamics in deep neural networks (DNNs) trained with stochastic gradient descent (SGD) and study the effect of learning rate $\eta$, depth $d$, and width $w$ of the neural network. By analyzing the maximum eigenvalue $\lambda_t^H$ of the Hessian of the loss, which is a measure of sharpness of the loss landscape, we find that the dynamics can show four distinct regimes: (i) an early time transient regime, (ii) an intermediate saturation regime, (iii) a progressive sharpening regime, and (iv) a late time "edge of stability" regime. The early and intermediate regimes (i) and (ii) exhibit a rich phase diagram depending on $\eta \equiv c/\lambda_0^H$, $d$, and $w$. We identify several critical values of $c$, which separate qualitatively distinct phenomena in the early time dynamics of training loss and sharpness. Notably, we discover the opening up of a "sharpness reduction" phase, where sharpness decreases at early times, as $d$ and $1/w$ are increased.

## 1 Introduction

The optimization dynamics of deep neural networks (DNNs) is a rich problem that is of great interest. Basic questions about how to choose learning rates and their effect on generalization error and training speed remain intensely studied research problems. Classical intuition from convex optimization has lead to the often made suggestion that in stochastic gradient descent (SGD), the learning rate $\eta$ should satisfy $\eta < 2/\lambda^H$, where $\lambda^H$ is the maximum eigenvalue of the Hessian $H$ of the loss, in order to ensure that the network reaches a minimum. However several recent studies have suggested that it is both possible and potentially preferable to have the learning rate *early in training* reach $\eta > 2/\lambda^H$ [66, 49, 72]. The idea is that such a choice will induce a temporary training instability, causing the network to 'catapult' out of a local basin into a flatter one with lower $\lambda^H$ where training stabilizes. Indeed, during the early training phase, the local curvature of the loss landscape changes rapidly [42, 1, 37, 16], and the learning rate plays a crucial role in determining the convergence basin [37]. Flatter basins are believed to be preferable because they potentially lead to lower generalization error [31, 32, 42, 12, 39, 14] and allow larger learning rates leading to potentially faster training.

From a different perspective, the major theme of deep learning is that it is beneficial to increase the model size as much as possible. This has come into sharp focus with the discovery of scaling laws that show power law improvement in generalization error with model and dataset size [40]. This raises the fundamental question of how one can scale DNNs to arbitrarily large sizes while maintaining the ability to learn; in particular, how should initialization and optimization hyperparameters be chosen to maintain a similar quality of learning as the model size is taken to infinity [34, 47, 48, 11, 69, 58, 70, 68]?

---

[*]Condensed Matter Theory Center, University of Maryland, College Park

[†]Institute for Physical Science and Technology, University of Maryland, College Park

[‡]Department of Physics, University of Maryland, College Park

[§]Joint Quantum Institute, University of Maryland, College Park

37th Conference on Neural Information Processing Systems (NeurIPS 2023).

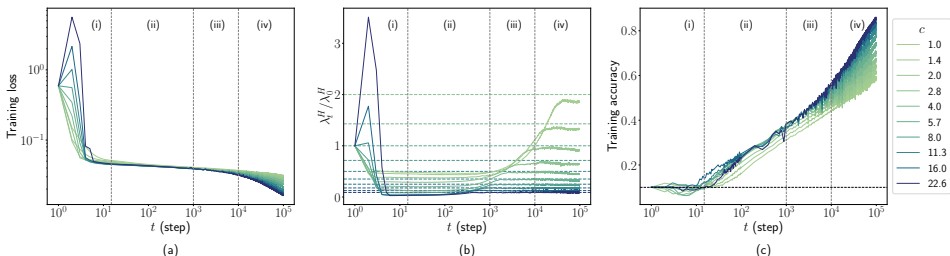

Figure 1: Training trajectories of the (a) training loss, (b) sharpness, and (c) training accuracy of CNNs ($d = 5$ and $w = 512$) trained on CIFAR-10 with MSE loss using vanilla SGD with learning rates $\eta = c/\lambda_0^H$ and batch size $B = 512$. Vertical dashed lines approximately separate the different training regimes. Horizontal dashed lines in (b) denote the $2/\eta$ threshold for each learning rate.

Motivated by these ideas, we perform a systematic analysis of the training dynamics of SGD for DNNs as learning rate, depth, and width are tuned, across a variety of architectures and datasets. We monitor both the loss and sharpness ($\lambda^H$) trajectories during early training, observing a number of qualitatively distinct phenomena summarized below.

## 1.1 Our contributions

We study SGD on fully connected networks (FCNs) with the same number of hidden units (width) in each layer, convolutional neural networks (CNNs), and ResNet architectures of varying width $w$ and depth $d$ with ReLU activation. For CNNs, the width corresponds to the number of channels. We focus on networks parameterized in Neural Tangent Parameterization (NTP) [34], and Standard Parameterization (SP) [62] initialized at criticality [55, 58], while other parameterizations and initializations may show different behavior. Further experimental details are provided in Appendix A. We study both mean-squared error (MSE) and cross-entropy loss functions and the datasets CIFAR-10, MNIST, Fashion-MNIST. Our findings apply to networks with $d/w \lesssim C$, where $C$ depends on architecture class (e.g. for FCNs, $C \approx 1/16$) and loss function, but is independent of $d$, $w$, and $\eta$. Above this ratio, the dynamics becomes noise-dominated, and separating the underlying deterministic dynamics from random fluctuations becomes challenging, as shown in Appendix E. We use sharpness to refer to $\lambda_t^H$, the maximum eigenvalue of $H$ at time-step $t$, and flatness refers to $1/\lambda_t^H$.

By monitoring the sharpness, we find four clearly separated, qualitatively distinct regimes throughout the training trajectory. Fig. 1 shows an example from a CNN architecture. The four observed regimes are: (i) an early time transient regime where loss and sharpness may drastically change and eventually settle down, (ii) an intermediate saturation regime where the sharpness has lowered and remains relatively constant, (iii) a progressive sharpening regime where sharpness steadily rises, and finally, (iv) a late time regime where the sharpness saturates around $2/\eta$ for MSE loss; whereas for cross-entropy loss, sharpness drops after reaching this maximum value while remaining less than $2/\eta$ [8]. Note the log scale in Figure 1 highlights the early regimes (i) and (ii); in absolute terms these are much shorter in time than regimes (iii) and (iv).

In this work, we focus on the early transient and intermediate saturation regimes. As learning rate, $d$ and $w$ are tuned, a clear picture emerges, leading to a rich phase diagram, as demonstrated in Section 2. Given the learning rate scaled as $\eta = c/\lambda_0^H$, we characterize four distinct behaviors in the training dynamics in the early transient regime (i):

**Sharpness reduction phase** ($c < c_{loss}$) **:** *Both the loss and the sharpness* monotonically decrease during early training. There is a particularly significant drop in sharpness in the regime $c_{crit} < c < c_{loss}$, which motivates us to refer to learning rates lower than $c_{crit}$ as sub-critical and larger than $c_{crit}$ as super-critical. We discuss $c_{crit}$ in detail below. The regime $c_{crit} < c < c_{loss}$ opens up significantly with increasing $d$ and $1/w$, which is a new result of this work.

**Loss catapult phase** ($c_{loss} < c < c_{sharp}$) **:** The first few gradient steps take training to a flatter region but with a higher loss. Training eventually settles down in the flatter region as the loss starts to decrease again. The sharpness *monotonically decreases from initialization* in this early time transient regime.

**Loss and sharpness catapult phase** ($c_{sharp} < c < c_{max}$)**:** In this regime *both the loss and sharpness* initially start to increase, effectively catapulting to a different point where loss and sharpness can start to decrease again. Training eventually exhibits a significant reduction in sharpness by the end of the early training. The report of a *loss and sharpness catapult* is also new to this work.

**Divergent phase** ($c > c_{max}$)**:** The learning rate is too large for training and the loss diverges.

The critical values $c_{loss}$, $c_{sharp}$, $c_{max}$ are random variables that depend on random initialization, SGD batch selection, and architecture. The averages of $c_{loss}$, $c_{sharp}$, $c_{max}$ shown in the phase diagrams show strong systematic dependence on depth and width. In order to better understand the cause of the sharpness reduction during early training we study the effect of network output at initialization by (1) centering the network, (2) setting last layer weights to zero, or (3) tuning the overall scale of the output layer. We also analyze the linear connectivity of the loss landscape in the early transient regime and show that for a range of learning rates $c_{loss} < c < c_{barrier}$, no barriers exist from the initial state to the final point of the initial transient phase, even though training passes through regions with higher loss than initialization.

Next, we provide a quantitative analysis of the intermediate saturation regime. We find that sharpness during this time typically displays 3 distinct regimes as the learning rate is tuned, depicted in Fig. 5. By identifying an appropriate order parameter, we can extract a sharp peak corresponding to $c_{crit}$. For MSE loss $c_{crit} \approx 2$, whereas for crossentropy loss, $4 \gtrsim c_{crit} \gtrsim 2$. For $c \ll c_{crit}$, the network is effectively in a lazy training regime, with increasing fluctuations as $d$ and/or $1/w$ are increased.

Finally, we show that a single hidden layer linear network – the $uv$ model – displays the same phenomena discussed above and we analyze the phase diagram in this minimal model.

## 1.2 Related works

A significant amount of research has identified various training regimes using diverse criteria, e.g., [13, 1, 15, 37, 17, 45, 35, 8, 33]. Here we focus on studies that characterize training regimes with sharpness and learning rates. Several studies have analyzed sharpness at different training times [37, 16, 35, 8, 33]. Ref. [8] studied sharpness at late training times and showed how *large-batch* gradient descent shows progressive sharpening followed by the edge of stability, which has motivated various theoretical studies [9, 2, 3]. Ref. [37] studied the entire training trajectory of sharpness in models trained with SGD and cross-entropy loss and found that sharpness increases during the early stages of training, reaches a peak, and then decreases. In contrast, we find a sharpness-reduction phase, $c < c_{loss}$ which becomes more prominent with increasing $d$ and $1/w$, where sharpness only decreases during early training; this also occurs in the catapult phase $c_{loss} < c < c_{sharp}$, during which the loss initially increases before decreasing. This discrepancy is likely due to different initialization and learning rate scaling in their work [33].

Ref. [35] examined the effect of hyperparameters on sharpness at late training times. Ref. [20] studied the optimization dynamics of SGD with momentum using sharpness. Ref. [45] classify training into 2 different regimes using training loss, providing a significantly coarser description of training dynamics than provided here. Ref. [33] studied the scaling of the maximum learning rate with $d$ and $w$ during early training in FCNs and its relationship with sharpness at initialization.

Refs. [52, 71] present phase diagrams of shallow ReLU networks at infinite width under gradient flow. Previous studies such as [41, 69] show that $2/\lambda_0^{NTK}$ is the maximum learning rate for convergence as $w \to \infty$. This limit results in the kernel regime as training time is restricted to $O(1)$ in the limit of infinite width, resulting in a lazy training regime for learning rates less than $2/\lambda_0^{NTK}$ and divergent training for larger learning rates. In contrast, we analyze optimization dynamics at training timescales $t_\star$ that grow with width $w$. Specifically, the end of the early time transient period occurs at $t_\star \sim \log(w)$.

Ref. [49] analyzed the training dynamics at large widths and training times, using the top eigenvalue of the neural tangent kernel (NTK) as a proxy for sharpness. They demonstrated the existence of a new early training phase, which they dubbed the "catapult" phase, $2/\lambda_0^{NTK} < \eta < \eta_{max}$, in wide networks trained with MSE loss using SGD, in which training converges after an initial increase in training loss. The existence of this new training regime was further extended to quadratic models with large widths by [72, 53]. Our work extends the above analysis by studying the combined effect of learning rate, depth, and width for both MSE and cross-entropy loss, demonstrating the opening

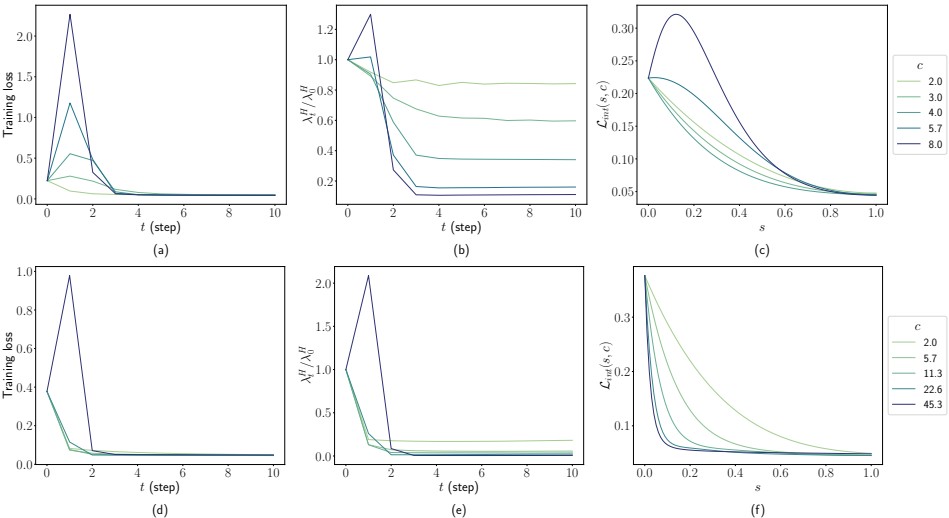

Figure 2: Early training dynamics of (a, b, c) a shallow ($d = 5$, $w = 512$) and (d, e, f) a deep CNN ($d = 10, w = 128$) trained on CIFAR-10 with MSE loss for $t = 10$ steps using SGD for various learning rates $\eta = c/\lambda_0^H$ and batch size $B = 512$. (a, d) training loss, (b, e) sharpness, and (c, f) interpolated loss between the initial and final parameters after 10 steps for the respective models. For the shallow CNN, $c_{loss} = 2.82, c_{sharp} = 5.65, c_{max} = 17.14$ and for the deep CNN, $c_{loss} = 36.75, c_{sharp} = 39.39, c_{max} = 48.50$.

of a sharpness-reduction phase, the refinement of the catapult phase into two phases depending on whether the sharpness also catapults, analyzing the phase boundaries as $d$ and $1/w$ is increased, analyzing linear mode connectivity in the catapult phase, examining different qualitative behaviors in the intermediate saturation regime (ii) mentioned above.

## 2 Phase diagram of early transient regime

For wide enough networks trained with MSE loss using SGD, training converges into a flatter region after an initial increase in the training loss for learning rates $c > 2$ [49]. Fig. 2(a, b) shows the first 10 steps of the loss and sharpness trajectories of a shallow ($d = 5$ and $w = 512$) CNN trained on the CIFAR-10 dataset with MSE loss using SGD. For learning rates, $c \geq 2.82$, the loss catapults and training eventually converges into a flatter region, as measured by sharpness. Additionally, we observe that sharpness may also spike initially, similar to the training loss (see Fig. 2 (b)). However, this initial spike in sharpness occurs at relatively higher learning rates ($c \geq 5.65$), which we will examine along with the loss catapult. We refer to this spike in sharpness as 'sharpness catapult.'

An important consideration is the degree to which this phenomenon changes with network depth and width. Interestingly, we found that the training loss in deep networks on average catapults at much larger learning rates than $c = 2$. Fig. 2(d, e) shows that for a deep ($d = 10, w = 128$) CNN, the loss and sharpness may catapult only near the maximum trainable learning rate. In this section, we characterize the properties of the early training dynamics of models with MSE loss. In Appendix F, we show that a similar picture emerges for cross-entropy loss, despite the dynamics being noisier.

### 2.1 Loss and sharpness catapult during early training

In this subsection, we characterize the effect of finite depth and width on the onset of the loss and sharpness catapult and training divergence. We begin by defining critical constants that correspond to the above phenomena.

**Definition 1.** ($c_{loss}, c_{sharp}, c_{max}$) *For learning rate $\eta = c/\lambda_0^H$, let the training loss and sharpness at step $t$ be denoted by $\mathcal{L}_t(c)$ and $\lambda_t^H(c)$. We define $c_{loss}(c_{sharp})$ as minimum learning rates constants*

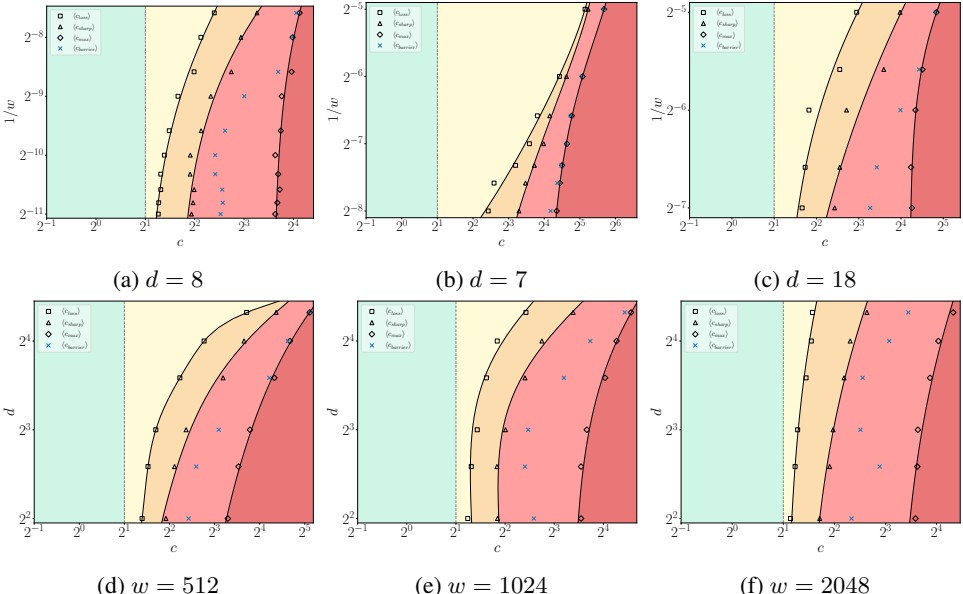

| (a) $d = 8$ | (b) $d = 7$ | (c) $d = 18$ |
| (d) $w = 512$ | (e) $w = 1024$ | (f) $w = 2048$ |

Figure 3: Phase diagrams of early training of neural networks trained with MSE loss using SGD. Panels (a-c) show phase diagrams with width: (a) FCNs ($d = 8$) trained on the MNIST dataset, (b) CNNs ($d = 7$) trained on the Fashion-MNIST dataset, (c) ResNet ($d = 18$) trained on the CIFAR-10 (without batch normalization). Panels (d-f) show phase diagrams with depth: FCNs trained on the Fashion-MNIST dataset for different widths. Each data point in the figure represents an average of ten distinct initializations, and the solid lines represent a smooth curve fitted to the raw data points. The vertical dotted line shows $c = 2$ for comparison, and various colors are filled in between the various curves for better visualization. For experimental details and additional results, see Appendices A and C, respectively. The phase diagram of early training of FCNs with depth for three different widths trained on Fashion-MNIST with MSE loss using SGD.

*such that the loss (sharpness) increases during the initial transient period:*

$$c_{loss} = \min_c \{c \mid \max_{t \in [1, T_1]} \mathcal{L}_t(c) > \mathcal{L}_0(c)\}, \qquad c_{sharp} = \min_c \{c \mid \max_{t \in [1, T_1]} \lambda_t^H(c) > \lambda_0^H(c)\},$$

*and $c_{max}$ as the maximum learning rate constant such that the loss does not diverge during the initial transient period: $c_{max} = \max_c \{c \mid \mathcal{L}_t(c) < K, \forall t \in [1, T_1]\}$, where $K$ is a fixed large constant.*[5]

Note that the definition of $c_{max}$ allows for more flexibility than previous studies [33] in order to investigate a wider range of phenomena occurring near the maximum learning rate. Here, $c_{loss}$, $c_{sharp}$, and $c_{max}$ are random variables that depend on the random initialization and the SGD batch sequence, and we denote the average over this randomness using $\langle \cdot \rangle$.

Fig. 3(a-c) illustrates the phase diagram of early training for three different architectures trained on various datasets with MSE loss using SGD. These phase diagrams show how the averaged values $\langle c_{loss} \rangle$, $\langle c_{sharp} \rangle$, and $\langle c_{max} \rangle$ are affected by width. The results show that the averaged values of all the critical constants increase significantly with $1/w$ (note the log scale). At large widths, the loss starts to catapult at $c \approx 2$. As $1/w$ increases, $\langle c_{loss} \rangle$ increases and eventually converges to $\langle c_{max} \rangle$ at large $1/w$. By comparison, sharpness starts to catapult at relatively large learning rates at small $1/w$, with $\langle c_{sharp} \rangle$ continuing to increase with $1/w$ while remaining between $\langle c_{loss} \rangle$ and $\langle c_{max} \rangle$. Similar results are observed for different depths as demonstrated in Appendix C. Phase diagrams obtained by varying $d$ are qualitatively similar to those obtained by varying $1/w$, as shown in Figure 3(d-f). Comparatively, we observe that $\langle c_{max} \rangle$ may increase or decrease with $1/w$ in different settings while consistently increasing with $d$, as shown in Appendices F and H.

---

[5]We use $K = 10^5$ to estimate $c_{max}$ In all our experiments, $\mathcal{L}_0 = \mathcal{O}(1)$ (see Appendix A), which justifies the use of a fixed value.

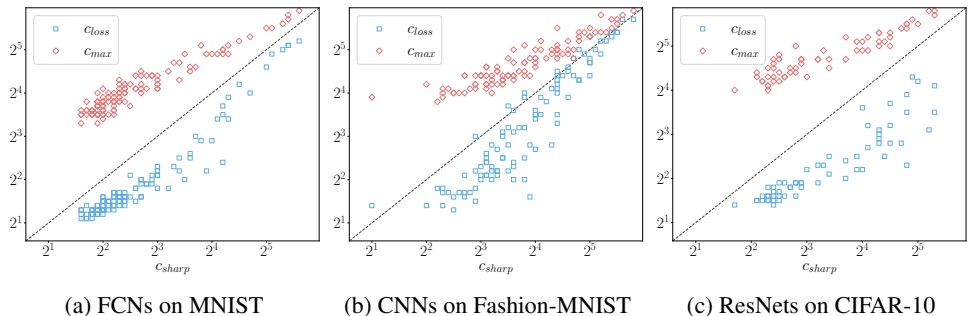

(a) FCNs on MNIST    (b) CNNs on Fashion-MNIST    (c) ResNets on CIFAR-10

Figure 4: The relationship between critical constants for (a) FCNs, (b) CNNs, and (c) ResNets. Each data point corresponds to a run with varying depth, width, and initialization. The dashed line represents the $y = x$ line.

While we plotted the averaged quantities $\langle c_{loss}\rangle$, $\langle c_{sharp}\rangle$, $\langle c_{max}\rangle$, we have observed that their variance also increases significantly with $d$ and $1/w$; in Appendix C we show standard deviations about the averages for different random initializations. Nevertheless, we have found that the inequality $c_{loss} \leq c_{sharp} \leq c_{max}$ typically holds, for any given initialization and batch sequences, except for some outliers due to high fluctuations when the averaged critical curves start merging at large $d$ and $1/w$. Fig. 4 shows evidence of this claim. The setup is the same as in Fig. 3. Appendix D presents extensive additional results across various architectures and datasets.

In Appendix F, we show that cross-entropy loss shows similar results with some notable differences. The loss catapults at a relatively higher value $\langle c_{loss}\rangle \gtrsim 4$ and $\langle c_{max}\rangle$ consistently decreases with $1/w$, while still satisfying $c_{loss} \leq c_{sharp} \leq c_{max}$.

## 2.2 Loss connectivity in the early transient period

In the previous subsection, we observed that training loss and sharpness might quickly increase before decreasing ("catapult") during early training for a range of depths and widths. A logical next step is to analyze the region in the loss landscape that the training reaches after the catapult. Several works have analyzed loss connectivity along the training trajectory [21, 51, 64]. Ref. [51] report that training traverses a barrier at large learning rates, aligning with the naive intuition of a barrier between the initial and final points of the loss catapult, as the loss increases during early training. In this section, we will test the credibility of this intuition in real-world models. Specifically, we linearly interpolate the loss between the initial and final point after the catapult and examine the effect of the learning rate, depth, and width. The linearly interpolated loss and barrier are defined as follows.

**Definition 2.** $(\mathcal{L}_{int}(s, c), U(c))$ *Let $\theta_0$ represent the initial set of parameters, and let $\theta_{T_1}$ represent the set of parameters at the end of the initial transient period, trained using a learning rate constant $c$. Then, we define the linearly interpolated loss as $\mathcal{L}_{int}(s, c) = \mathcal{L}[(1 - s)\, \theta_0 + s\, \theta_{T_1}]$, where $s \in [0, 1]$ is the interpolation parameter. The interpolated loss barrier is defined as the maximum value of the interpolated loss over the range of $s$: $U(c) = \max_{s\in[0,1]} \mathcal{L}_{int}(s) - \mathcal{L}(\theta_0)$.*

Here we subtracted the loss's initial value such that a positive value indicates a barrier to the final point from initialization. Using the interpolated loss barrier, we define $c_{barrier}$ as follows.

**Definition 3.** $(c_{barrier})$ *Given the initial ($\theta_0$) and final parameters ($\theta_{T_1}$), we define $c_{barrier}$ as the minimum learning rate constant such that there exists a barrier from $\theta_0$ to $\theta_{T_1}$: $c_{barrier} = \min_c\{c \mid U(c) > 0\}$.*

Here, $c_{barrier}$ is also a random variable that depends on the initialization and SGD batch sequence. We denote the average over this randomness using $\langle .\rangle$ as before. Fig. 2(c, f) shows the interpolated loss of CNNs trained on the CIFAR-10 dataset for $t = 10$ steps. The experimental setup is the same as in Section 2. For the network with larger width, we observe a barrier emerging at $c_{barrier} = 5.65$, while the loss starts to catapult at $c_{loss} = 2.83$. In comparison, we do not observe any barrier from initialization to the final point at large $d$ and $1/w$. Fig. 3 shows the relationship between $\langle c_{barrier}\rangle$ and $1/w$ for various models and datasets. We consistently observe that $c_{sharp} \leq c_{barrier}$, suggesting that training traverses a barrier only when sharpness starts to catapult during early training. Similar results

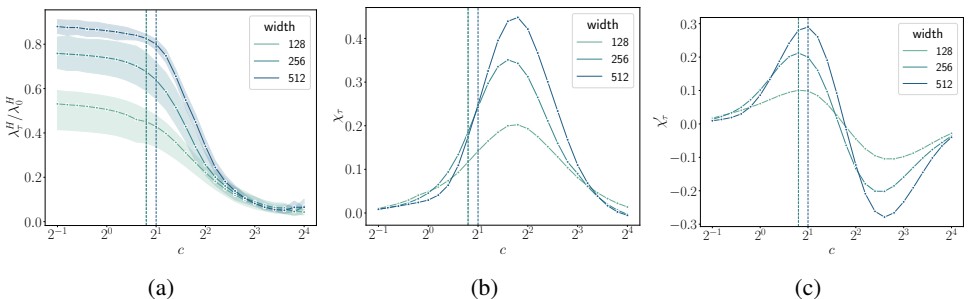

(a)  (b)  (c)

Figure 5: (a) Normalized sharpness measured at $c\tau = 200$ against the learning rate constant for 7-layer CNNs trained on the CIFAR-10 dataset, with varying widths. Each data point is an average over 5 initializations, where the shaded region depicts the standard deviation around the mean trend. (b, c) Smooth estimations of the first two derivatives, $\chi_\tau$ and $\chi'_\tau$, of the averaged normalized sharpness wrt the learning rate constant. The vertical lines denote $c_{crit}$ estimated using the maximum of $\chi'_\tau$. For smoothening details, see Appendix I.2.

were observed on increasing $d$ instead of $1/w$ as shown in Appendix C. We chose not to characterize the phase diagram of early training using $c_{barrier}$ as we did for other critical $c$'s, as it is somewhat different in character than the other critical constants, which depend only on the sharpness and loss trajectories.

These observations call into question the intuition of catapulting out of a basin for a range of learning rates in between $c_{loss} < c < c_{barrier}$. These results show that for these learning rates, the final point after the catapult already lies in the same basin as initialization, and even *connected through a linear path*, revealing an inductive bias of the training process towards regions of higher loss during the early time transient regime.

## 3  Intermediate saturation regime

In the intermediate saturation regime, sharpness does not change appreciably and reflects the cumulative change that occurred during the initial transient period. This section analyzes sharpness in the intermediate saturation regime by studying how it changes with the learning rate, depth, and width of the model. Here, we show results for MSE loss, whereas cross-entropy results are shown in Appendix F.

We measure the sharpness $\lambda_\tau^H$ at a time $\tau$ in the middle of the intermediate saturation regime. We choose $\tau$ so that $c\tau \approx 200$.[6] For further details on sharpness measurement, see Appendix I.1. Fig. 5(a) illustrates the relationship between $\lambda_\tau^H$ and the learning rate for 7-layer deep CNNs trained on the CIFAR-10 dataset with varying widths. The results indicate that the dependence of $\lambda_\tau^H$ on learning rate can be grouped into three distinct stages. (1) At small learning rates, $\lambda_\tau^H$ remains relatively constant, with fluctuations increasing as $d$ and $1/w$ increase ($c < 2$ in Fig. 5(a)). (2) A crossover regime where $\lambda_\tau^H$ is dropping significantly ($2 < c < 2^3$ in Fig. 5(a)). (3) A saturation stage where $\lambda_\tau^H$ stays small and constant with learning rate ($c > 2^3$) in Fig. 5(a). In Appendix I, we show that these results are consistent across architectures and datasets for varying values of $d$ and $w$. Additionally, the results reveal that in stage (1), where $c < 2$ is sub-critical, $\lambda_\tau^H$ decreases with increasing $d$ and $1/w$. In other words, for small $c$ and in the intermediate saturation regime, the loss is locally flatter as $d$ and $1/w$ increase.

We can precisely extract a critical value of $c$ that separates stages (1) and (2), which corresponds to the onset of an abrupt reduction of sharpness $\lambda_\tau^H$. To do this, we consider the averaged normalized sharpness over initializations and denote it by $\langle \lambda_\tau^H / \lambda_0^H \rangle$. The first two derivatives of the averaged normalized sharpness, $\chi_\tau = -\frac{\partial}{\partial c} \langle \lambda_\tau^H / \lambda_0^H \rangle$ and $\chi'_\tau = -\frac{\partial^2}{\partial c^2} \langle \lambda_\tau^H / \lambda_0^H \rangle$, characterize the change in sharpness with learning rate. The extrema of $\chi'_\tau$ quantitatively define the boundaries between the three stages described above. In particular, using the maximum of $\chi'_\tau$, we define $\langle c_{crit} \rangle$, which marks the beginning of the sharp decrease in $\lambda_\tau^H$ with the learning rate.

---

[6]time-step $\tau = 200/c$ is in the middle of regime (ii) for the models studied. Normalizing by $c$ allows proper comparison for different learning rates.

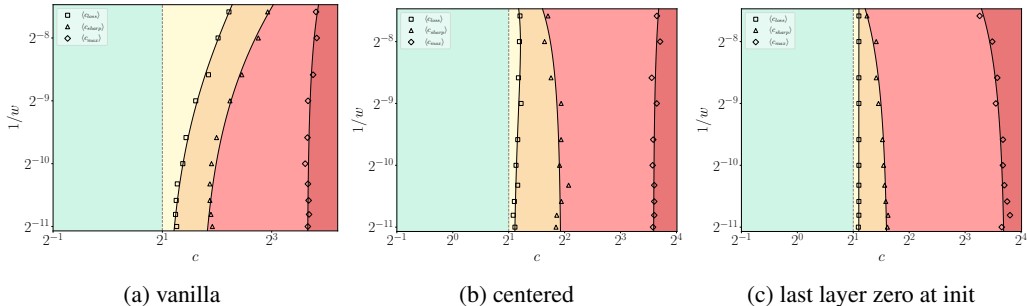

| (a) vanilla | (b) centered | (c) last layer zero at init |

Figure 6: Phase diagrams of $d = 8$ layer FCNs trained on the CIFAR-10 dataset using MSE, demonstrating the effect of output scale at initialization: (a) vanilla network, (b) centered network, and (c) network initialized with the last layer set to zero.

**Definition 4.** *($\langle c_{crit} \rangle$) Given the averaged normalized sharpness $\langle \lambda_\tau^H / \lambda_0^H \rangle$ measured at $\tau$, we define $c_{crit}$ to be the learning rate constant that minimizes its second derivative: $\langle c_{crit} \rangle = \arg\max_c \chi_\tau'$.*

Here, we use $\langle . \rangle$ to denote that the critical constant is obtained from the averaged normalized sharpness. Fig. 5(b, c) show $\chi_\tau$ and $\chi_\tau'$ obtained from the results in Fig. 5(a). We observe similar results across various architectures and datasets, as shown in Appendix I. Our results show that $\langle c_{crit} \rangle$ has slight fluctuations as $d$ and $1/w$ are changed but generally stay in the vicinity of $c = 2$. The peak in $\chi_\tau'$ becomes wider as $d$ and $1/w$ increase, indicating that the transition between stages (1) and (2) becomes smoother, presumably due to larger fluctuations in the properties of the Hessian $H$ at initialization. In contrast to $\langle c_{crit} \rangle$, $\langle c_{loss} \rangle$ increase with $d$ and $1/w$, implying the opening of the sharpness reduction phase $\langle c_{crit} \rangle < c < \langle c_{loss} \rangle$ as $d$ and $1/w$ increase. In Appendix F, we show that cross-entropy loss shows qualitatively similar results, but with $2 \lesssim \langle c_{crit} \rangle \lesssim 4$.

## 4 Effect of network output at initialization on early training

Here we discuss the effect of network output $f(x; \theta_t)$ at initialization on the early training dynamics. $x$ is the input and $\theta_t$ denotes the set of parameters at time $t$. We consider setting the network output to zero at initialization, $f(x; \theta_0) = 0$, by either (1) considering the "centered" network: $f_c(x; \theta) = f(x; \theta) - f(x; \theta_0)$, or (2) setting the last layer weights to zero at initialization (for details, see Appendix G). Remarkably, both (1) and (2) remove the opening up of the sharpness reduction phase with $1/w$ as shown in Figure 6. The average onset of the loss catapult, diagnosed by $\langle c_{loss} \rangle$, becomes independent of $1/w$ and $d$.

We also empirically study the impact of the output scale [19, 5, 4] on early training dynamics. Given a network function $f(x; \theta)$, we define the scaled network as $f_s(x; \theta) = \alpha f(x; \theta)$, where $\alpha$ is a scalar, fixed throughout training. In Appendix H, we show that a large (resp. small) value of $\|f(x; \theta_0)\|$ relative to the one-hot encodings of the labels causes the sharpness to decrease (resp. increase) during early training. Interestingly, we still observe an increase in $\langle c_{loss} \rangle$ with $d$ and $1/w$, unlike the case of initializing network output to zero, highlighting the unique impact of output scale on the dynamics.

## 5 Insights from a simple model

Here we analyze a two-layer linear network [56, 60, 49], the $uv$ model, which shows much of the phenomena presented above. Define $f(x) = \frac{1}{\sqrt{w}} v^T u x$, with $x, f(x) \in \mathbb{R}$. Here, $u, v \in \mathbb{R}^w$ are the trainable parameters, initialized using the normal distribution, $u_i, v_i \sim \mathcal{N}(0, 1)$ for $i \in \{1, \ldots, w\}$. The model is trained with MSE loss on a single training example $(x, y) = (1, 0)$, which simplifies the loss to $\mathcal{L}(u, v) = f^2/2$, and which was also considered in Ref. [49]. Our choice of $y = 0$ is motivated by the results of Sec. 4, which suggest that the empirical results of Sec. 2 are intimately related to the model having a large initial output scale $\|f(x; \theta_0)\|$ relative to the output labels. We minimize the loss using gradient descent (GD) with learning rate $\eta$. The early time phase diagram also shows similar features to those described in preceding sections (compare Fig. 7(a) and Fig. 3). Below we develop an understanding of this early time phase diagram in the $uv$ model.

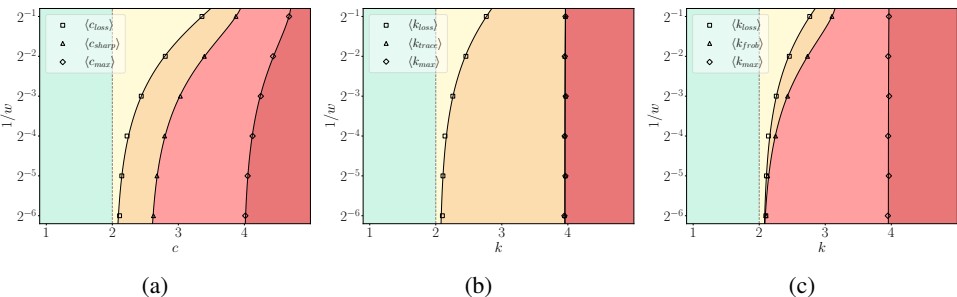

(a)                    (b)                    (c)

Figure 7: The phase diagram of the $uv$ model trained with MSE loss using gradient descent with (a) the top eigenvalue of Hessian $\lambda_t^H$, (b) the trace of Hessian $\text{tr}(H_t)$ and (c) the square of the Frobenius norm $\text{tr}(H_t^T H_t)$ used as a measure of sharpness. In (a), the learning rate is scaled as $\eta = c/\lambda_0^H$, while in (b) and (c), the learning rate is scaled as $\eta = k/\text{tr}(H_0)$. The vertical dashed line shows $c = 2$ ($k = 2$) for reference. Each data point is an average over $500$ random initializations.

The update equations of the $uv$ model in function space can be written in terms of the trace of the Hessian $\text{tr}(H)$

$$f_{t+1} = f_t \left(1 - \eta \,\text{tr}(H_t) + \frac{\eta^2 f_t^2}{w}\right), \qquad \text{tr}(H_{t+1}) = \text{tr}(H_t) + \frac{\eta f_t^2}{w}\left(\eta \,\text{tr}(H_t) - 4\right). \quad (1)$$

From the above equations, it is natural to scale the learning rate as $\eta = k/\text{tr}(H_0)$. Note that $c = \eta \lambda_0^H = k\lambda_0^H/\text{tr}(H_0)$. Also, we denote the critical constants in this scaling as $k_{loss}$, $k_{trace}$, $k_{max}$ and $k_{crit}$, where the definitions follow from Definitions 1 and 4 on replacing sharpness with trace and use $\langle.\rangle$ to denote an average over initialization. Figure 7(b) shows the phase diagram of early training, with $\text{tr}(H_t)$ replaced with $\lambda_t^H$ as the measure of sharpness and with the learning rate scaled as $\eta = k/\text{tr}(H_0)$. Similar to Figure 7(a), we observe a new phase $\langle k_{crit}\rangle < k < \langle k_{loss}\rangle$ opening up at small width. However, we do not observe the loss-sharpness catapult phase as $\text{tr}(H)$ does not increase during training (see Equation 1). We also observe $\langle k_{max}\rangle = 4$, independent of width.

In Appendix B.3, we show that the critical value of $k$ for which $\langle \mathcal{L}_1/\mathcal{L}_0 \rangle > 1$ increases with $1/w$, which explains why $\langle k_{loss}\rangle$ increases with $1/w$. Combined with $\langle k_{crit}\rangle \approx 2$, this implies the opening up of the sharpness reduction phase as $w$ is decreased.

To understand the loss-sharpness catapult phase, we require some other measure as $\text{tr}(H)$ does not increase for $0 < k < 4$. As $\lambda_t^H$ is difficult to analyze, we consider the Frobenius norm $\|H\|_F = \sqrt{\text{tr}(H^T H)}$ as a proxy for sharpness. We define $k_{frob}$ as the minimum learning rate such that $\|H_t\|_F^2$ increases during early training. Figure 7(c) shows the phase diagram of the $uv$ model, with $\|H_t\|_F^2$ as the measure of sharpness, while the learning rate is scaled as $\eta = k/\text{tr}(H_0)$. We observe the loss-sharpness catapult phase at small widths. In Appendix B.4, we show that the critical value of $k$ for which $\langle \|H_1\|_F^2 - \|H_0\|_F^2 \rangle > 0$ increases from $\langle k_{loss}\rangle$ as $1/w$ increases. This explains the opening up of the loss catapult phase at small $w$ in Fig. 7 (c).

Fig. 8 shows the training trajectories of the $uv$ model with large ($w = 512$) and small ($w = 2$) widths in a two-dimensional slice of parameters defined by $\text{tr}(H)$ and weight correlation $\langle v, u \rangle/\|u\|\|v\|$. The above figure reveals that the first few training steps of the small-width network take the system in a flatter direction (as measured by $\text{tr}(H)$) as compared to the wider network. This means that the small-width network needs a relatively larger learning rate to get to a point of increased loss (loss catapult). We thus have the opening up of a new regime $\langle k_{crit}\rangle < k < \langle k_{loss}\rangle$, in which the loss and sharpness monotonically decrease during early training.

The loss landscape of the $uv$ model shown in Fig. 8 reveals interesting insights into the loss landscape connectivity results in Section 2.2 and the presence of $c_{barrier}$. Fig. 8 shows how even when there is a loss catapult, as long as the learning rate is not too large, the final point after the catapult can be reached from initialization by a linear path without increasing the loss and passing through a barrier. However if the learning rate becomes large enough, then the final point after the catapult may correspond to a region of large weight correlation, and there will be a barrier in the loss upon linear interpolation.

The $uv$ model trained on an example $(x, y)$ with $y \neq 0$ provides insights into the effect of network output at initialization observed in Section 4. In Appendix G, we show that setting $f_0 = 0$ and



Figure 8: Training trajectories of the $uv$ model trained on $(x, y) = (1, 0)$, with (a, b) large and (c, d) small width, in a two-dimensional slice of the parameters defined by the trace of Hessian $\mathrm{tr}(H)$ and weight correlation, trained with (a, c) small ($c = 0.5$) and (b, d) large ($c = 2.5$) learning rates. The colors correspond to the training loss $\mathcal{L}$, with darker colors representing a smaller loss.

$y \neq 0$ in the dynamical equations results in loss catapult at $k = 2$, implying $\langle k_{loss} \rangle \approx \langle k_{crit} \rangle \approx 2$, irrespective of $w$.

## 6   Discussion

We have studied the effect of learning rate, depth, and width on the early training dynamics in DNNs trained using SGD with learning rate scaled as $\eta = c/\lambda_0^H$. We analyzed the early transient and intermediate saturation regimes and presented a rich phase diagram of early training with learning rate, depth, and width. We report two new phases, sharpness reduction and loss-sharpness catapult, which have not been reported previously. Furthermore, we empirically investigated the underlying cause of sharpness reduction during early training. Our findings show that setting the network output to zero at initialization effectively leads to the vanishing of sharpness reduction phase at supercritical learning rates. We further studied loss connectivity in the early transient regime and demonstrated the existence of a regime $\langle c_{loss} \rangle < c < \langle c_{barrier} \rangle$, in which the final point after the catapult lies in the same basin as initialization, connected through a linear path. Finally, we study these phenomena in a 2-layer linear network ($uv$ model), gaining insights into the opening of the sharpness reduction phase.

We performed a preliminary analysis on the effect of batch size on the presented results in Appendix J. The sharpness trajectories of models trained with a smaller batch size ($B = 32$ vs. $B = 512$) show similar early training dynamics. In the early transient regime, we observe a qualitatively similar phase diagram. In the intermediate saturation regime, the effect of reducing the batch size is to broaden the transition around $c_{crit}$.

In Section 2, we noted that for cross-entropy loss, the loss starts to catapult around $c \approx 4$ at large widths, as compared to $c_{loss} = 2$ for MSE loss. Previous work, such as [50], analyzed the catapult dynamics for the $uv$ model with logistic loss and demonstrated that the loss catapult occurs above $\eta_{loss} = 4/\lambda_0^{NTK}$. We summarize the main intuition about their analysis in Appendix B.9. However, a complete understanding of the catapult phenomenon in the context of cross-entropy loss requires a more detailed examination.

The early training dynamics is sensitive to the initialization scheme and optimization algorithm used, and we leave it to future work to explore this dependence and its implications. In this work, we focused on models initialized at criticality [55] as it allows for proper gradient flow through ReLU networks at initialization [23, 58], and studied vanilla SGD for simplicity. However, other initializations [46], parameterizations [69, 70], and optimization procedures [22] may show dissimilarities with the reported phase diagram of early training.

## Acknowledgments

We thank Andrey Gromov, Tianyu He, and Shubham Jain for discussions, and Paolo Glorioso, Sho Yaida, Daniel Roberts, and Darshil Doshi for detailed comments on the manuscript. We also express our gratitude to anonymous reviewers for their valuable feedback for improving the manuscript. This work is supported by an NSF CAREER grant (DMR1753240) and the Laboratory for Physical Sciences through the Condensed Matter Theory Center.

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

# A Experimental details

**Datasets:** We considered the MNIST [10], Fashion-MNIST [67], and CIFAR-10 [44] datasets. We standardized the images and used one-hot encoding for the labels.

**Models:** We considered fully connected networks (FCNs), Myrtle family CNNs [61] and ResNets (version 1) [29] trained using the JAX [7], and Flax libraries [30]. We use $d$ and $w$ to denote the depth and width of the network. Below, we provide additional details of the models and clarify what width corresponds to for CNNs and ResNets.

1. **FCNs:** We considered ReLU FCNs with constant width $w$ in Neural Tangent Parameterization (NTP) / Standard Parameterization (SP), initialized at criticality [55]. The models do not include bias or normalization. The forward pass of the pre-activations from layer $l$ to $l + 1$ is given by

$$h_i^{l+1} = \gamma^l \sum_j^w W_{ij}^l \phi(h_j^l),\tag{2}$$

where $\phi(.)$ is the ReLU activation and $\gamma^l$ is a constant. For NTP, $\gamma^l = 2/\sqrt{w}$ and the weights $W^l$ are initialized using normal distribution, i.e., $W_{ij}^l \sim \mathcal{N}(0, 1)$. For SP, $\gamma^l = 1$ and the weights $W^l$ are initialized as $W_{ij}^l \sim \mathcal{N}(0, 2/w)$. For the last layer, we have $\gamma^L = 1/\sqrt{w}$ for NTP and $W_{ij}^L \sim \mathcal{N}(0, 1/w)$ for SP.

   For $d/w \gtrsim 1/16$, the dynamics is noisier, and it becomes challenging to separate the underlying deterministic dynamics from random fluctuations (see Appendix E).

2. **CNNs:** We considered Myrtle family ReLU CNNs [61] without any bias or normalization in Standard Parameterization (SP), initialized using He initialization [29]. The above model uses a fixed number of channels in each layer, which we refer to as the width of the network. In this case, the forward pass equations for the pre-activations from layer $l$ to layer $l + 1$ are given by

$$h_i^{l+1}(\alpha) = \sum_j^w \sum_{\beta \in ker} W_{ij}^{l+1}(\beta) \phi(h_i^l(\alpha + \beta)),\tag{3}$$

where $\alpha, \beta$ label the spacial location. The weights are initialized as $W_{ij}^l(\beta) \sim \mathcal{N}(0, 2/k^2 w)$, where $k$ is the filter size.

3. **ResNets:** We considered version 1 ResNet [29] implementations from Flax examples without Batch Norm or regularization. For ResNets, width corresponds to the number of channels in the first block. For example, the standard ResNet-18 has four blocks with widths $[w, 2w, 4w, 8w]$, with $w = 64$. We refer to $w$ as the width or the widening factor. We considered ResNet-18 and ResNet-34.

All the models are trained with the average loss over the batch $\mathcal{D}_B = \{(x_\mu, y_\mu)\}_{\mu=1}^B$, i.e., $L(x, y_{\mathcal{D}_B}) = 1/B \sum_{\mu=1}^B \ell(x_\mu, y_\mu)$, where $\ell(.)$ is the loss function. This normalization, along with initialization, ensures that the loss is $\mathcal{O}(1)$ at initialization.

**Bias:** Throughout this work, we have primarily focused on models without any bias for simplicity. In Appendix K, we demonstrate that bias does not have an appreciable impact on the results.

**Batch size:** We use a batch size of 512 and scale the learning rate as $\eta = c/\lambda_0^H$ in all our experiments, unless specified. Appendix J shows results for a smaller batch size $B = 32$.

**Learning rate:** We scale the learning rate constant as $c = 2^x$, with $x \in \{-1.0, \ldots x_{max}\}$ in steps of 0.1. Here, $x_{max}$ is related to the maximum learning rate constant as $c_{max} = 2^{x_{max}}$.

**Sharpness measurement:** We measure sharpness using the power iteration method with 20 iterations. We found that 20 iterations suffice both for MSE and cross-entropy loss. For MSE loss, we use $m = 2048$ randomly selected training examples for evaluating sharpness at each step. In comparison, we found that cross-entropy requires a large number of training examples to obtain a good approximation

of sharpness. Given the computational constraints, we use $4096$ training examples to approximate sharpness for cross-entropy loss.

**Averages over initialization and SGD runs:** All the critical constants depend on both the random initializations and the SGD runs. In our experiments, we found that the fluctuations from initialization at large $d/w$ outweigh the randomness coming from different SGD runs. Thus, we focus on initialization averages in all our experiments.

## A.1 Compute usage

We utilized different computational resources depending on the task complexity. For less demanding tasks, we performed computation for a total of $2800$ hours, utilizing a seventh of an NVIDIA A100 GPU. For more computationally intensive tasks, we utilized a full NVIDIA A100 GPU for a total $300$ hours.

## A.2 Reproducibility

The main results of this paper can be reproduced using the associated GitHub repository:https://github.com/dayal-kalra/early-training.

## A.3 Details of Figures in the main text:

**Figure 1:** A shallow CNN ($d = 5$, $w = 128$) in SP trained on the CIFAR-10 dataset with MSE loss for 1000 epochs using SGD with learning rates $\eta = c/\lambda_0^H$ and batch size $B = 512$. We measure sharpness at every step for the first epoch, every epoch between 10 and 100 epochs, and every 10 epochs beyond 100.

**Figure 2:** (top panel) A wide ($d = 5$, $w = 512$) and (bottom panel) a deep CNN ($d = 10, w = 128$) in SP trained on the CIFAR-10 dataset with MSE loss for $t = 10$ steps using vanilla SGD with learning rates $\eta = c/\lambda_0^H$ and batch size $B = 512$.

**Figure 3:** Phase diagrams of early training of neural networks trained with MSE loss using SGD. Panels (a-c) show phase diagrams with width: (a) FCNs ($d = 8$) trained on the MNIST dataset, (b) CNNs ($d = 7$) trained on the Fashion-MNIST dataset, (c) ResNet ($d = 18$) trained on the CIFAR-10 (without batch normalization). Panels (d-f) show phase diagrams with depth: FCNs trained on the Fashion-MNIST dataset for different widths. Each data point in the figure represents an average of ten distinct initializations, and the solid lines represent a two-degree polynomial $y = a + bx + cx^2$ fitted to the raw data points. Here, where $x = 1/w$, and $y$ can take on one of three values: $c_{loss}, c_{sharp}$ and $c_{max}$.

**Figure 4:** (a) FCNs in SP with $d \in \{4, 8, 16\}$ and $w \in \{256, 512, 1024, 2048\}$trained on the MNIST dataset, (b) CNNs in SP with $d \in \{5, 7, 10\}$ and $w \in \{64, 128, 256, 512\}$ trained on the Fashion-MNIST dataset, (c) ResNet in SP with $d \in \{18, 34\}$ and $w \in \{32, 64, 128\}$ trained on the CIFAR-10 dataset (without batch normalization).

**Figure 6:** Phase diagrams of $d = 8$ layer FCNs trained on the CIFAR-10 dataset using MSE, demonstrating the effect of output scale at initialization: (a) vanilla network, (b) centered network, and (c) network initialized with the last layer set to zero. The values of widths are the same as in Figure 3.

**Figure 5:** Normalized sharpness measured at $c\tau = 200$ against the learning rate constant for 7-layer CNNs in SP trained on the CIFAR-10 dataset, with $w \in \{128, 256, 512\}$. Each data point is an average over five random initialization. Smoothening details are provided in Appendix I.2.

**Figure 7:** The phase diagram of the $uv$ model trained with MSE loss using gradient descent with (a) the top eigenvalue of Hessian $\lambda_t^H$, (b) the trace of Hessian $\text{tr}(H_t)$ and (c) the square of the Frobenius norm $\text{tr}(H_t^T H_t)$ used as a measure of sharpness. In (a), the learning rate is scaled as $\eta = c/\lambda_0^H$, while in (b) and (c), the learning rate is scaled as $\eta = k/\text{tr}(H_0)$. The vertical dashed line shows $c = 2$ ($k = 2$) for reference. Each data point is an average over 500 random initializations.

**Figure 8:** Training trajectories of the $uv$ model with (a, b) large ($w = 512$) and (c, d) small ($w = 2$) width, trained for $t = 10$ training steps on a single example $(x, y) = (1, 0)$ with MSE loss using vanilla gradient descent with learning rates (a, c) $c = 0.5$ and (b, d) $c = 2.50$.

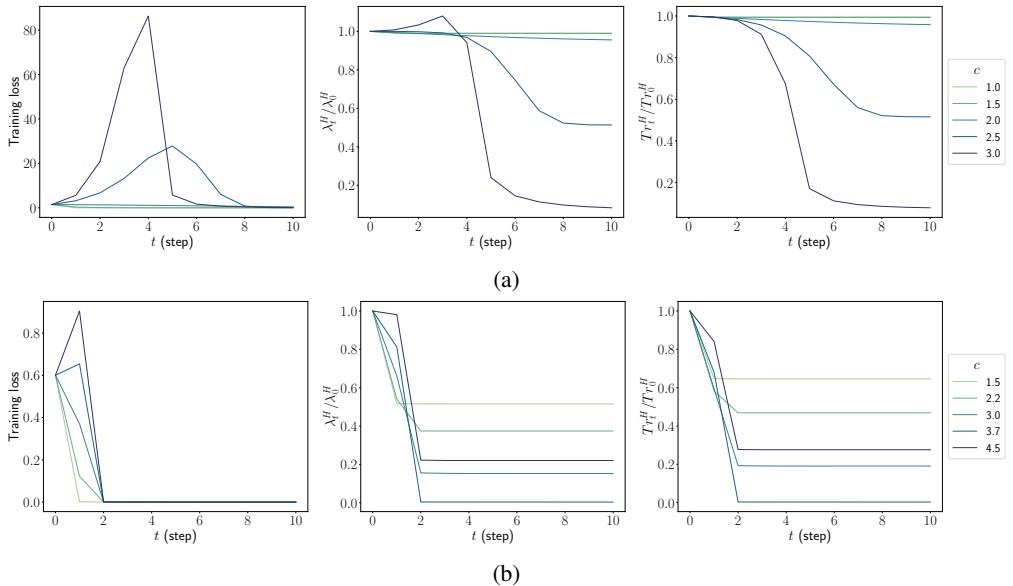

Figure 9: Training trajectories of the $uv$ model with (a) large ($w = 512$) and (v) a small ($w = 2$) widths trained for $t = 10$ training steps on a single example $(x, y) = (1, 0)$. For the wide network, $c_{loss} = 2.1$, $c_{sharp} = 2.6$, $c_{max} = 4.0$, and for the narrow network, $c_{loss} = 3.74$, $c_{sharp} = 4.63$, $c_{max} = 4.93$.

## B  Additional results for the $uv$ model

### B.1  Details of the model

Consider a two-layer linear network in (NTP) with unit input-output dimensions

$$f(x) = \frac{1}{\sqrt{w}} v^T u x, \tag{4}$$

where $x, f(x) \in \mathbb{R}$. Here, $u, v \in \mathbb{R}^w$ are trainable parameters, with each element initialized using the normal distribution, $u_i, v_i \sim \mathcal{N}(0, 1)$ for $i \in \{1, \ldots, w\}$. The model is trained using MSE loss on a single training example $(x, y) = (1, 0)$, which simplifies the loss to

$$\mathcal{L}(u, v) = \frac{1}{2} f^2. \tag{5}$$

The trace of the Hessian $\operatorname{tr}(H)$ has a simple expression in terms of the norms of the weight vectors

$$\operatorname{tr}(H) = \frac{x^2}{w} \left( \|u\|^2 + \|v\|^2 \right), \tag{6}$$

which is equivalent to the NTK for this model. The Frobenius norm of the Hessian $\|H\|_F$ can be written in terms of the loss $\mathcal{L}$ and $\operatorname{tr}(H)$

$$\|H\|_F^2 = \operatorname{tr}(H)^2 + 2f^2 \left( 1 + \frac{2}{w} \right) = \operatorname{tr}(H)^2 + 4\mathcal{L} \left( 1 + \frac{2}{w} \right) \tag{7}$$

The gradient descent updates of the model trained using MSE loss on a single training example $(x, y) = (1, 0)$ are given by

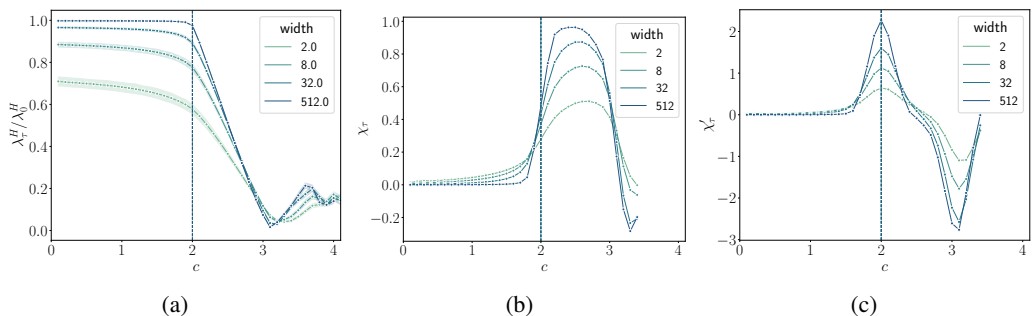

(a)        (b)        (c)

Figure 10: (a) Normalized sharpness measured at $\tau = 100$ steps against the learning rate constant for the $uv$ model trained on $(x, y) = (1, 0)$, with varying widths. Each data point is an average of over 500 initializations, where the shaded region depicts the standard deviation around the mean trend. (b, c) Smooth estimations of the first two derivatives, $\chi_\tau$ and $\chi'_\tau$, of the, averaged normalized sharpness wrt the learning rate constant. The vertical dashed lines denote $c_{crit}$ estimated for each width, using the maximum of $\chi'_\tau$. Here, we have removed the points beyond $c = 3.5$ for the calculation of derivatives to avoid large fluctuations near the divergent phase. Smoothening details are described in Appendix I.2.

$$v_{t+1} = v_t - \eta f_t \frac{1}{\sqrt{w}} u_t x \tag{8}$$

$$u_{t+1} = u_t - \eta f_t \frac{1}{\sqrt{w}} v_t x \tag{9}$$

The update equations in function space can be written in terms of the trace of the Hessian $\mathrm{tr}(H)$.

$$
\begin{aligned}
f_{t+1} &= f_t \left( 1 - \eta\, \mathrm{tr}(H_t) + \frac{\eta^2 f_t^2}{w} \right) \\
\mathrm{tr}(H_{t+1}) &= \mathrm{tr}(H_t) + \frac{\eta f_t^2}{w} \left( \eta\, \mathrm{tr}(H_t) - 4 \right).
\end{aligned}
\tag{10}
$$

Figure 9 shows the training trajectories of the $uv$ model trained on $(x, y) = (1, 0)$ using MSE loss for 10 training steps. The model shows similar dynamics to those presented in Section 2. It is worth mentioning that the above equations have been analyzed in [49] at large width. In the following subsections, we extend their analysis by incorporating the higher-order terms to analyze the effect of finite width.

## B.2 The intermediate saturation regime

The $uv$ model trained on $(x, y) = (1, 0)$ does not show the progressive sharpening and late-time regimes (iii) and (iv) described in Section 1. Hence, we can measure sharpness at the end of training to analyze how it is reduced upon increasing the learning rate and to compare it with the intermediate saturation regime results in Section 3.

Figure 10(a) shows the normalized sharpness measured at $\tau = 100$ steps for various widths. This behavior reproduces the results observed in the intermediate saturation regime in Section 3. In particular, we can see stages (1) and (2), where $\lambda_\tau^H / \lambda_0^H$ starts off fairly independent of learning rate constant $c$, and then dramatically reduces when $c > 2$; stage (3), where $\lambda_\tau^H / \lambda_0^H$ plateaus at a small value as a function of $c$ is too close to the divergent phase in this model to be clearly observed. The corresponding derivatives of the averaged normalized sharpness, $\chi_\tau$, and $\chi'_\tau$, are shown in Figure 10(b, c). The vertical dashed lines denote $c_{crit}$ estimated for each width, using the maximum of $\chi'_\tau$. We observe that $c_{crit} = 2$ for all widths.

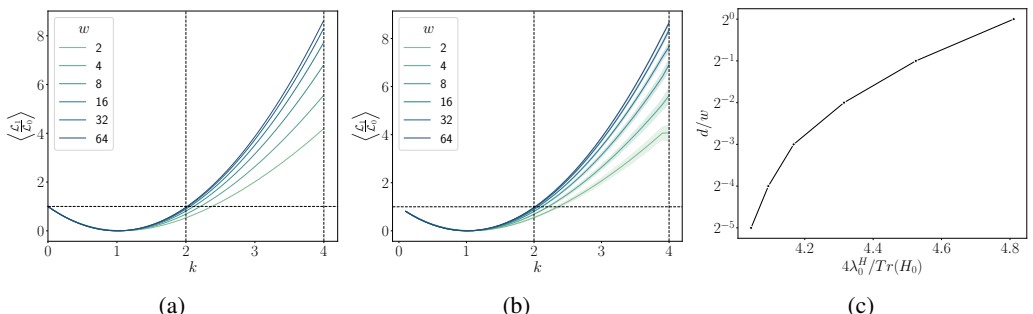

(a)            (b)            (c)

Figure 11: (a, b) The averaged loss at the first step $\langle \mathcal{L}_1/\mathcal{L}_0 \rangle$ against the learning rate constant $k$ for varying widths obtained from (a) inequality 16 and (b) numerical experiments. The intersection of $\langle \mathcal{L}_1/\mathcal{L}_0 \rangle$ with the horizontal line $y = 1$ depicts $k_{loss}$. The two vertical lines $k = 2$ and $k = 4$ mark the endpoints of $k_{loss}$ at small and large widths. The shaded region in (b) shows the standard deviation around the mean trend. (c) The scaling of $\lambda_0^H$ and $\mathrm{tr}(H_0)$ with width.

### B.3   Opening of the sharpness reduction phase in the $uv$ model

This section shows that $\mathcal{O}(1/w)$ terms in Equation (10) effectively lead to the opening of the sharpness reduction phase with $1/w$ in the $uv$ model. In Appendix B.2, we demonstrated that for the $uv$ model, $c_{crit} = 2$ for all values of widths. Hence, it suffices to show that $c_{loss}$ increases from the value 2 as $1/w$ increases. We do so by finding the smallest $k$ such that the averaged loss over initializations increases during early training.

It follows from Equation 10 that the averaged loss increases in the first training step if the following holds

$$\left\langle \frac{\mathcal{L}_1}{\mathcal{L}_0} \right\rangle = \left\langle \left( 1 - \eta \, \mathrm{tr}(H_0) + \frac{\eta^2 f_0^2}{w} \right)^2 \right\rangle > 1, \tag{11}$$

where $\langle . \rangle$ denotes the average over initializations. On scaling the learning rate with trace as $\eta = k/\mathrm{tr}(H_0)$, we have

$$\left\langle \frac{\mathcal{L}_1}{\mathcal{L}_0} \right\rangle = \left\langle \left( 1 - k + \frac{k^2}{w} \frac{f_0^2}{\mathrm{tr}(H_0)^2} \right) \right\rangle > 1 \tag{12}$$

$$\left\langle \frac{\mathcal{L}_1}{\mathcal{L}_0} \right\rangle = \left( (1-k)^2 + 2(1-k) \frac{k^2}{w} \left\langle \frac{f_0^2}{\mathrm{tr}(H_0)^2} \right\rangle + \frac{k^4}{w^2} \left\langle \frac{f_0^4}{\mathrm{tr}(H_0)^4} \right\rangle \right) > 1. \tag{13}$$

The required two averages have the following expressions as shown in Appendix B.8.

$$\left\langle \frac{f_0^2}{Tr(H_0)^2} \right\rangle = \frac{w}{4(w+1)} \tag{14}$$

$$\left\langle \frac{f_0^4}{Tr(H_0)^4} \right\rangle = \frac{3(w+2)w^3}{16} \frac{\Gamma(w)}{\Gamma(w+4)}. \tag{15}$$

Inserting the above expressions in Equation 13, on average the loss increases in the very first step if the following inequality holds

$$\left\langle \frac{\mathcal{L}_1}{\mathcal{L}_0} \right\rangle = \left( (1-k)^2 + \frac{k^2(1-k)}{2(w+1)} + \frac{3k^4}{16(w+3)(w+1)} \right) > 1 \tag{16}$$

The graphical representation of the above inequality shown in Figure 11(a) is in excellent agreement with the experimental results presented in Figure 11(b).

Let us denote $k'_{loss}$ as the minimum learning rate constant such that the average loss increases in the first step. Similarly, let $k_{loss}$ denote the learning rate constant if the loss increases in the first 10 steps. Then, $k'_{loss}$ increases from the value 2 as $1/w$ increases as shown in Figure 11(a). By comparison, the trace reduces at any step if $\eta \operatorname{tr}(H_t) < 4$. At initialization, this condition becomes $k < 4$. Hence, for $k < k'_{loss}$, both the loss and trace monotonically decrease in the first training step. These arguments can be extended to later training steps, revealing that the loss and trace will continue to decrease for $k < k'_{loss}$.

Next, let $\eta_{loss}$ denote the learning rate corresponding to $c_{loss}$. Then, we have $\eta_{loss} = \frac{c_{loss}}{\lambda_0^H} = \frac{k_{loss}}{\operatorname{tr}(H_0)}$, implying

$$c_{loss} = k_{loss} \frac{\lambda_0^H}{\operatorname{tr}(H_0)}. \tag{17}$$

Figure 11(c) shows that $\lambda_0^H \geq \operatorname{tr}(H_0)$ for all widths, implying $c_{loss} \geq k_{loss}$. Hence, $c_{loss}$ increases with $1/w$ as observed in Figure 7(a). In Appendix B.2, we demonstrated that for the $uv$ model, $c_{crit} = 2$ for all values of widths. Incorporating this with $c_{loss}$ increases with $1/w$, we have sharpness reduction phase opening up as $1/w$ increases.

### B.4 Opening of the loss catapult phase at finite width

In this section, we use the Frobenius norm of the Hessian $\|H\|_F$ as a proxy for the sharpness and demonstrate the emergence of the loss-sharpness catapult phase at finite width. In particular, We analyze the expectation value $\langle \operatorname{tr}(H^T H) \rangle$ after the first training step near $k = k_{loss}$ and show that $k_{loss} \leq k_{frob}$, with the difference increasing with $1/w$. First, we write $\operatorname{tr}(H_t^T H_t)$ in terms of $\mathcal{L}_t$ and $\operatorname{tr}(H_t)$

$$\operatorname{tr}(H_t^T H_t) = \operatorname{tr}(H_t)^2 + 4\left(1 + \frac{2}{w}\right)\mathcal{L}_t. \tag{18}$$

Next, using Equations 1, we write down the change in $\operatorname{tr}(H_t^T H_t)$ after the first training step in terms of $\operatorname{tr}(H_0)$ and $\mathcal{L}_0$

$$\Delta \operatorname{tr}(H_1^T H_1) = \operatorname{tr}(H_1^T H_1^T) - \operatorname{tr}(H_0^T H_0) = \operatorname{tr}(H_1)^2 - \operatorname{tr}(H_0)^2 + 4\left(1 + \frac{2}{w}\right)(\mathcal{L}_1 - \mathcal{L}_0)$$

$$\Delta \operatorname{tr}(H_1^T H_1) = \frac{\eta f_0^2}{w}(\eta \operatorname{tr}(H_0) - 4)\left[\frac{\eta f_0^2}{w}(\eta \operatorname{tr}(H_0) - 4) + 2\operatorname{tr}(H_0)\right] + 4\left(1 + \frac{2}{w}\right)(\mathcal{L}_1 - \mathcal{L}_0) \tag{19}$$

Next, we substitute $\eta = k/\operatorname{tr}(H_0)$ to obtain the above equation as a function of $k$

$$\Delta \operatorname{tr}(H_1^T H_1) = \frac{k(k-4)}{w}\left[\frac{k(k-4)}{w}\frac{f_0^4}{\operatorname{tr}(H_0)^2} + 2f_0^2\right] + 4\left(1 + \frac{2}{w}\right)(\mathcal{L}_1 - \mathcal{L}_0) \tag{20}$$

Finally, we calculate the expectation value of $\langle \Delta \operatorname{tr}(H_1^T H_1) \rangle$

$$\langle \Delta \operatorname{tr}(H_1^T H_1) \rangle = \frac{k(k-4)}{w}\left[\frac{k(k-4)}{w}\left\langle\frac{f_0^4}{\operatorname{tr}(H_0)^2}\right\rangle + 2\langle f_0^2\rangle\right] + 4\left(1 + \frac{2}{w}\right)\langle \mathcal{L}_1 - \mathcal{L}_0\rangle, \tag{21}$$

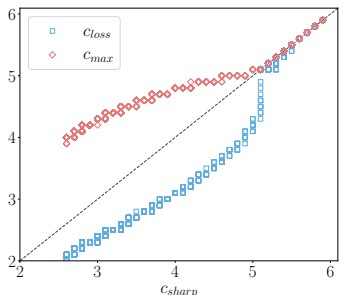

Figure 12: The relationship between the critical constants for the $uv$ model trained on a single training examples $(x, y) = (1, 0)$ with MSE loss using gradient descent. Each data point corresponds to a random initialization

by estimating $\left\langle \frac{f_0^4}{\text{tr}(H_0)^2} \right\rangle$ using the approach demonstrated in the previous section

$$\left\langle \frac{f_0^4}{\text{tr}(H_0)^2} \right\rangle = \frac{3w}{4(w+3)}. \tag{22}$$

Inserting $\left\langle \frac{f_0^4}{\text{tr}(H_0)^2} \right\rangle$ in Equation 21 along with $\langle f_0^2 \rangle = 1$, we have

$$\left\langle \Delta \text{tr}(H_1^T H_1) \right\rangle = \underbrace{\frac{k(k-4)}{w} \left[ \frac{3k(k-4)}{4(w+3)} + 2 \right]}_{I(k,w)} + 4\left(1 + \frac{2}{w}\right) \langle \mathcal{L}_1 - \mathcal{L}_0 \rangle \tag{23}$$

At infinite width, the above equation reduces to $\left\langle \Delta \text{tr}(H_1^T H_1) \right\rangle = 4 \langle \mathcal{L}_1 - \mathcal{L}_0 \rangle$, and hence, $k_{frob} = k_{loss}$. For any finite width, $I(k, w) < 0$ for $0 < k < 4$. At $k \leq k_{loss}$, $\mathcal{L}_1 - \mathcal{L}_0 \leq 0$, and therefore $\left\langle \Delta \text{tr}(H_1^T H_1) \right\rangle < 0$. In order for the sharpness to catapult, we require $\left\langle \Delta \text{tr}(H_1^T H_1) \right\rangle > 0$ and therefore $k_{frob} > k_{loss}$. As $1/w$ increases $|I(k, w)|$ also increases, which means a higher value of $\mathcal{L}_1 - \mathcal{L}_0$ is required to reach a point where $\left\langle \Delta \text{tr}(H_1^T H_1) \right\rangle \geq 0$. Thus $k_{frob} - k_{loss}$ increases with $1/w$.

### B.5 The early training trajectories

Figure 9 shows the early training trajectories of the $uv$ model with large ($w = 512$) and small ($w = 2$) widths. The dynamics depicted show several similarities with early training dynamics of real-world models shown in Figure 2. At small widths, the loss catapults at relatively higher learning rates (specifically, at $c_{loss} = 3.74$, which is significantly higher than the critical value of $c_{crit} = 2$).

### B.6 Relationship between critical constants

Figure 12 shows the relationship between various critical constants for the $uv$ model. The data show that the inequality $c_{loss} \leq c_{sharp} \leq c_{max}$ holds for every random initialization of the $uv$ model.

### B.7 Phase diagrams with error bars

This section shows the variation in the phase diagram boundaries of the $uv$ model shown in Figure 7(a, b). Figure 13 shows these phase diagrams. Each data point is an average of over 500 initializations. The horizontal bars around each data point indicate the region between 25% and 75% quantile.

### B.8 Derivation of the expectation values

Here, we provide the detailed derivation of the averages $\left\langle \frac{f_0^2}{Tr(H_0)^2} \right\rangle$ and $\left\langle \frac{f_0^4}{Tr(H_0)^4} \right\rangle$. We begin by finding the average $\left\langle \frac{f_0^2}{Tr(H_0)^2} \right\rangle$

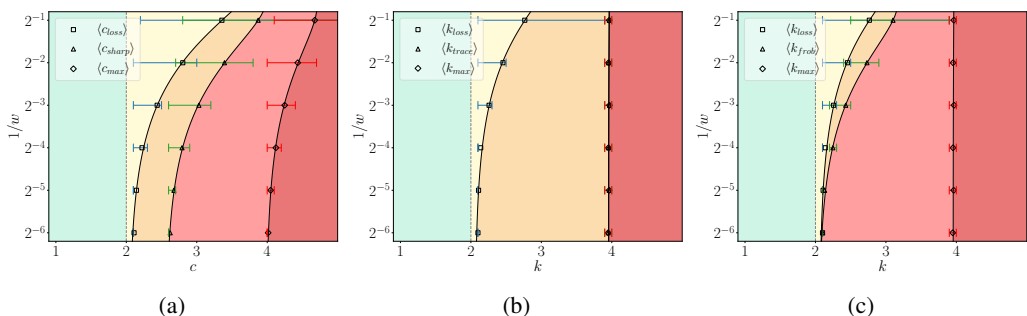

(a)               (b)               (c)

Figure 13: The phase diagram of the $uv$ model trained with MSE loss using gradient descent with (a) sharpness $\lambda_t^H$ (b) trace of Hessian $\mathrm{tr}_0^H$ and (c) the square of the Frobenius norm $\mathrm{tr}(H_t^T H_t)$ used as a measure of sharpness. In (a), the learning rate is scaled as $\eta = c/\lambda_0^H$, while in (b) and (c), the learning rate is scaled as $\eta = k/\mathrm{tr}(H_0)$. Each data point denotes an average of over $500$ initialization, and the smooth curve represents a 2-degree polynomial fitted to the raw data. The horizontal bars around the average data point indicate the region between $25\%$ and $75\%$ quantile.

$$\left\langle \frac{f_0^2}{Tr(H_0)^2} \right\rangle = w \int_{-\infty}^{\infty} \prod_{i=1}^{w} \left( \frac{dv_i du_i}{2\pi} \right) \exp\left( -\frac{\|u\|^2 + \|v\|^2}{2} \right) \frac{\sum_{j,k=1}^{w} u_j v_j u_k v_k}{\left( \|u\|^2 + \|v\|^2 \right)^2}, \quad (24)$$

where $\|.\|$ denotes the norm of the vectors.

The above integral is non-zero only if $j = k$. Hence, it is a sum of $w$ identical integrals. Without any loss of generality, we solve this integral for $j = 1$ and multiply by $w$ to obtain the final result, i.e.,

$$\left\langle \frac{f_0^2}{Tr(H_0)^2} \right\rangle = w^2 \int_{-\infty}^{\infty} \prod_{i=1}^{w} \left( \frac{dv_i du_i}{2\pi} \right) \exp\left( -\frac{\|u\|^2 + \|v\|^2}{2} \right) \frac{u_1^2 v_1^2}{\left( \|u\|^2 + \|v\|^2 \right)^2} \quad (25)$$

Consider a transformation of $u, v \in \mathbb{R}^w$ into $w$ dimensional spherical coordinates such that

$$u_1 = r_u \cos\varphi_{u_1}, \qquad\qquad v_1 = r_v \cos\varphi_{v_1}, \qquad (26)$$

which yields,

$$\left\langle \frac{f_0^2}{Tr(H_0)^2} \right\rangle = \frac{w^2}{(2\pi)^w} \int dr_u dr_v d\Omega_{u,w} d\Omega_{v,w} r_u^{w-1} r_v^{w-1} \exp\left( -\frac{r_u^2 + r_v^2}{2} \right) \frac{r_u^2 \cos^2\varphi_{u_1} r_v^2 \cos^2\varphi_{u_1}}{\left( r_u^2 + r_v^2 \right)^2}$$

$$(27)$$

$$\left\langle \frac{f_0^2}{Tr(H_0)^2} \right\rangle = \frac{w^2}{(2\pi)^w} \int dr_u dr_v \exp\left( -\frac{r_u^2 + r_v^2}{2} \right) \frac{r_u^2 r_v^2}{\left( r_u^{w+1} + r_v^{w+1} \right)^2} \int d\Omega_{u,w} d\Omega_{v,w} \cos^2\varphi_{u_1} \cos^2\varphi_{v_1}$$

$$(28)$$

$$\left\langle \frac{f_0^2}{Tr(H_0)^2} \right\rangle = \frac{w^2}{(2\pi)^w} \underbrace{\int dr_u dr_v \exp\left( -\frac{r_u^2 + r_v^2}{2} \right) \frac{r_u^2 r_v^2}{\left( r_u^{w+1} + r_v^{w+1} \right)^2}}_{I_r} \left( \underbrace{\int d\Omega_w \cos^2\varphi_1}_{I_\varphi} \right)^2,$$

$$(29)$$

where $d\Omega$ denotes the $w$ dimensional solid angle element. Here, we denote the radial and angular integrals by $I_r$ and $I_\varphi$ respectively. The radial integral $I_r$ is

$$I_r = \int_0^\infty dr_u dr_v \frac{r_u^{w+1} r_v^{w+1}}{(r_u^2 + r_v^2)^2} \exp\left(-\frac{r_u^2 + r_v^2}{2}\right). \tag{30}$$

Let $r_u = R\cos\theta$ and $r_v = R\sin\theta$ with $R \in [0,\infty)$ and $\theta \in [-\frac{\pi}{2}, \frac{\pi}{2}]$, then we have

$$I_r = \int_0^\infty dR\, R^{2w-1} e^{-R^2/2} \int_0^{\pi/2} d\theta \cos^{w+1}\theta \sin^{w+1}\theta \tag{31}$$

$$I_r = \frac{\sqrt{\pi}}{2^3} \frac{\Gamma(w)\, \Gamma\left(\frac{w+2}{2}\right)}{\Gamma\left(\frac{w+3}{2}\right)}, \tag{32}$$

where $\Gamma(.)$ denotes the Gamma function. The angular integral $I_\varphi$ is

$$I_\varphi = \int d\varphi_1 d\varphi_2 \ldots d\varphi_{w-1} \sin^{w-2}\varphi_1 \cos^2\varphi_1 \sin^{w-3}\varphi_2 \ldots \sin\varphi_{w-2} \tag{33}$$

$$I_\varphi = \int d\varphi_1 d\varphi_2 \ldots d\varphi_{w-1} \sin^{w-2}\varphi_1 \sin^{w-3}\varphi_2 \ldots \sin\varphi_{w-2} \frac{\int_0^\pi d\varphi_1 \sin^{w-2}\varphi_1 \cos^2\varphi_1}{\int_0^\pi d\varphi_1 \sin^{w-2}\varphi_1} \tag{34}$$

$$I_\varphi = \frac{\pi^{w/2}}{\Gamma(\frac{w+2}{2})}. \tag{35}$$

Plugging in Equations 32 and 35 into Equation 29, we obtain a very simple expression

$$\left\langle \frac{f_0^2}{Tr(H_0)^2} \right\rangle = \frac{w^2}{2^{w+3}} \frac{\sqrt{\pi}\Gamma(w)}{\Gamma(\frac{w+2}{2})\Gamma(\frac{w+3}{2})} = \frac{w}{4(w+1)}. \tag{36}$$

The other integral $\left\langle \frac{f_0^4}{Tr(H_0)^4} \right\rangle$ can be obtained by generalizing the above approach as described below

$$\left\langle \frac{f_0^4}{Tr(H_0)^4} \right\rangle = w^2 \int_{-\infty}^\infty \prod_{i=1}^w \left(\frac{dv_i du_i}{2\pi}\right) \exp\left(-\frac{\|u\|^2 + \|v\|^2}{2}\right) \frac{\sum_{j,k,l,m=1}^w u_j v_j u_k v_k u_l v_l u_m v_m}{(\|u\|^2 + \|v\|^2)^4}. \tag{37}$$

The integral is zero if either $j = k$ and $l = m$ or $j = k = l = m$, which we consider separately. Without loss of generality, we find the following integrals

$$\left\langle \frac{f_0^4}{Tr(H_0)^4} \right\rangle_{22} = w^2 \int_{-\infty}^\infty \prod_{i=1}^w \left(\frac{dv_i du_i}{2\pi}\right) \exp\left(-\frac{\|u\|^2 + \|v\|^2}{2}\right) \frac{u_1^2 u_2^2 v_1^2 v_2^2}{(\|u\|^2 + \|v\|^2)^4} \tag{38}$$

$$\left\langle \frac{f_0^4}{Tr(H_0)^4} \right\rangle_4 = w^2 \int_{-\infty}^\infty \prod_{i=1}^w \left(\frac{dv_i du_i}{2\pi}\right) \exp\left(-\frac{\|u\|^2 + \|v\|^2}{2}\right) \frac{u_1^4 v_1^4}{(\|u\|^2 + \|v\|^2)^4}, \tag{39}$$

which have the following expressions

$$\left\langle \frac{f_0^4}{Tr(H_0)^4} \right\rangle_{22} = \frac{w^2}{16} \frac{\Gamma(w)}{\Gamma(w+4)} \tag{40}$$

$$\left\langle \frac{f_0^4}{Tr(H_0)^4} \right\rangle_4 = \frac{9w^2}{16} \frac{\Gamma(w)}{\Gamma(w+4)}, \tag{41}$$

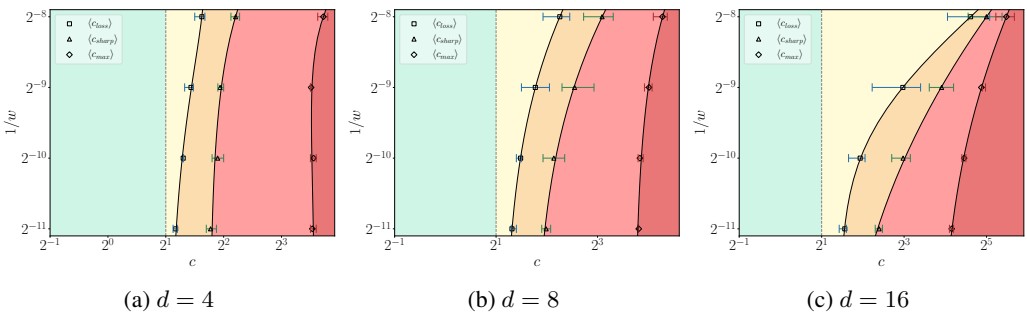

(a) $d = 4$       (b) $d = 8$       (c) $d = 16$

Figure 14: Phase diagrams of FCNs in NTP with varying depths trained on the MNIST dataset using MSE loss.

where $\Gamma(.)$ denotes the gamma function. On combining the expressions with their multiplicities, we obtain the final result

$$\left\langle \frac{f_0^4}{Tr(H_0)^4} \right\rangle = 3w(w-1) \left\langle \frac{f_0^4}{Tr(H_0)^4} \right\rangle_{22} + w \left\langle \frac{f_0^4}{Tr(H_0)^4} \right\rangle_{4} \tag{42}$$

$$\left\langle \frac{f_0^4}{Tr(H_0)^4} \right\rangle = \frac{3(w+2)w^3}{16} \frac{\Gamma(w)}{\Gamma(w+4)} \tag{43}$$

### B.9    Insights into the catapult effect in crossentropy loss using $uv$ model

In this section, we summarize the main intuition behind the discrepancy in the values of $c_{loss}$ for cross-entropy loss at large widths. We consider the $uv$ model trained on a classification task using logistic loss, as presented in [50].

Consider the $uv$ model trained on a binary classification task using the logistic loss on two training examples $(x_1, y_1) = (1, 1)$ and $(x_2, y_2) = (1, -1)$. Then, the total loss is $\mathcal{L}(f) = \frac{1}{2}\log(2 + 2\cosh(f))$. Hence, the loss grows monotonically as the output function $|f|$ increases. The update equation of the function is given by:

$$f_{t+1} = f_t \left( 1 - \frac{\eta \, tr(H_t)\mathcal{L}'(f)}{f_t} + \frac{\eta^2 \mathcal{L}'(f_t)^2}{n} \right), \tag{44}$$

where $\eta$ is the learning rate and $\mathcal{L}'(.)$ is the derivative of the loss. At large width, if the condition $|1 - \eta \, tr(H)\mathcal{L}'(f)/f| < 1$ holds, then output function continues to decrease. Given that $\mathcal{L}'(f)/f \leq 1/2$ in the above case, this decrease persists for $\eta \, tr(H) < 4$. This result provides some intuition behind the discrepancy.

## C    Phase diagrams of early training

This section describes experimental details and shows additional phase diagrams of early training. The results include (1) FCNs in NTP trained on MNIST, Fashion-MNIST, and CIFAR-10 datasets, (2) CNNs in SP trained on Fashion-MNIST and CIFAR-10, and (3) ResNets in SP trained on CIFAR-10 datasets using MSE loss. Figures 14 to 19 show these results. The depths and widths are the same as specified in Appendix A. Each data point is an average over 10 initializations. The horizontal bars around the average data point indicate the region between $25\%$ and $75\%$ quantile. Phase diagrams for cross-entropy results are shown in Appendix F.

**Additional experimental details**    : We train each model for $t = 10$ steps using SGD with learning rates $\eta = {}^c/\lambda_0^H$ and batch size of 512, where $c = 2^x$ with $x \in \{0.0, \ldots x_{max}\}$ in steps of 0.1. Here, $x_{max}$ is relatd to the maximum trainable learning rate constant as $c_{max} = 2^{x_{max}}$. We have considered 10 random initializations for each model. As mentioned in Appendix A, we do not

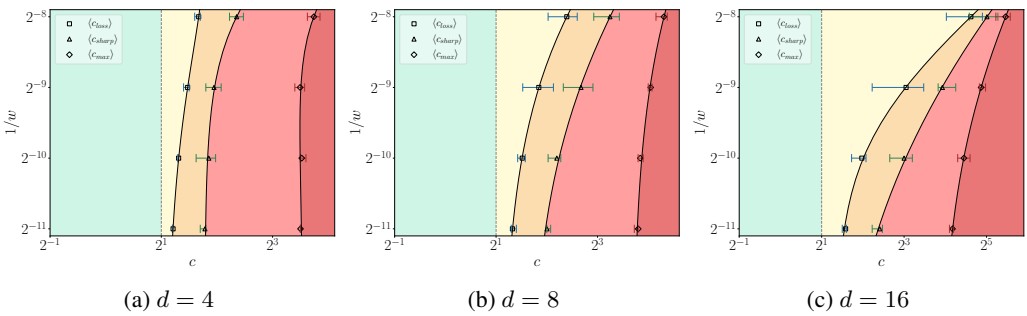

Figure 15: Phase diagrams of FCNs in NTP with varying depths trained on the Fashion-MNIST dataset.

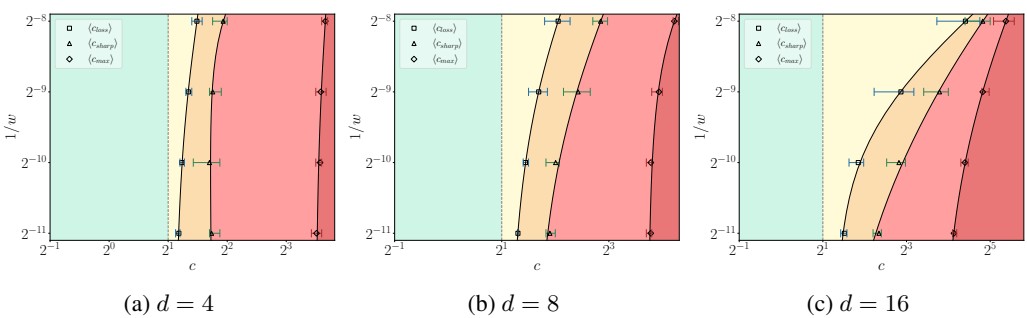

Figure 16: Phase diagrams of FCNs in NTP with varying depths trained on the CIFAR-10 dataset.

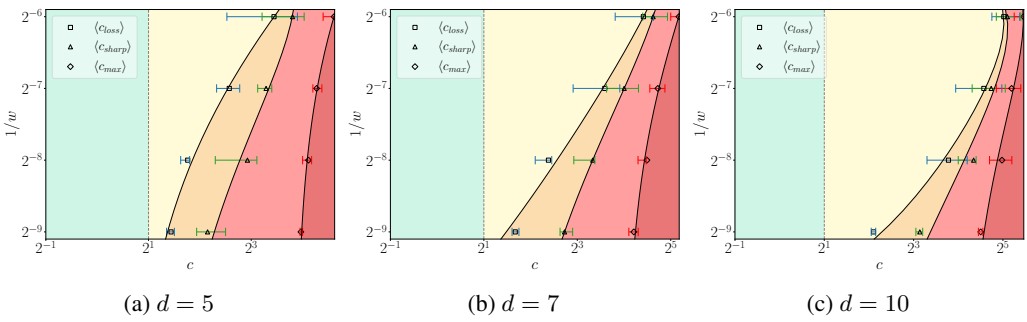

Figure 17: Phase diagrams of Convolutional Neural Networks (CNNs) in SP with varying depths trained on the Fashion-MNIST dataset.

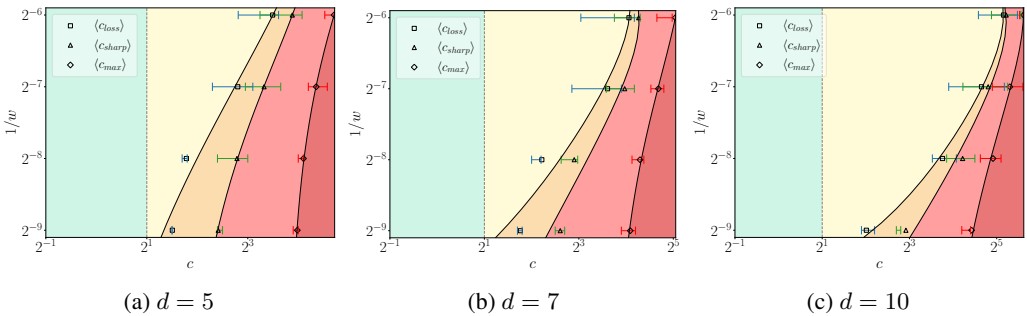

Figure 18: Phase diagrams of Convolutional Neural Networks (CNNs) in SP with varying depths trained on the CIFAR-10 dataset.

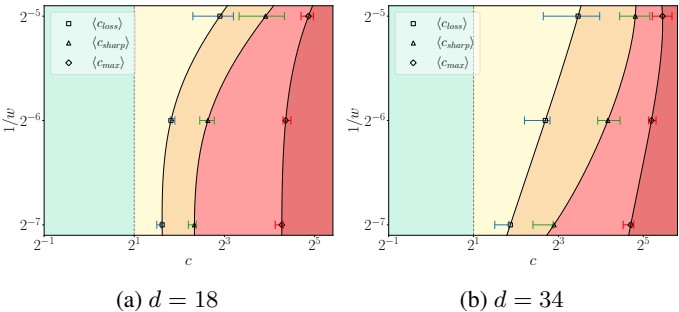

(a) $d = 18$              (b) $d = 34$

Figure 19: Phase diagrams of Resnets in SP with different depths trained on the CIFAR-10 dataset.

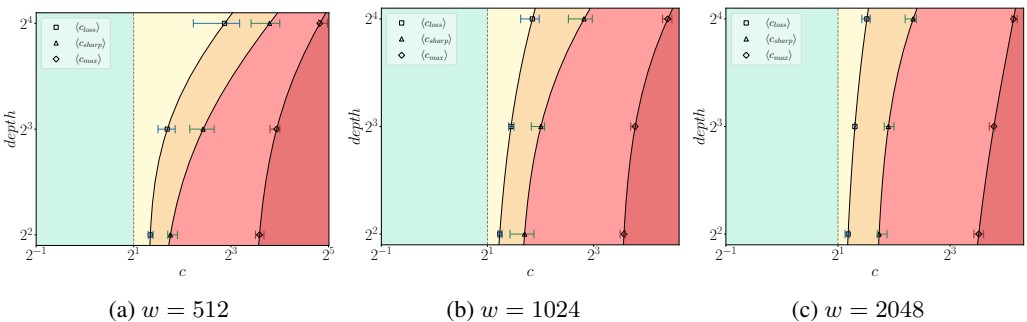

(a) $w = 512$       (b) $w = 1024$       (c) $w = 2048$

Figure 20: Phase diagrams of FCNs in NTP with varying widths trained on the CIFAR-10 dataset.

consider averages over SGD runs as the randomness from initialization outweighs it. Hence, we obtain 10 values for each of the critical values in the following results. For each initialization, we compute the critical constants using Definitions 1 and 3. To avoid a random increase in loss and sharpness due to fluctuations, we round off the values of $\lambda_t^H/\lambda_0^H$ and $\mathcal{L}_t/\mathcal{L}_0$ to their second decimal places before comparing with 1. We denote the average values using data points and variation using horizontal bars around the average data points, which indicate the region between $25\%$ and $75\%$ quantile. The smooth curves are obtained by fitting a two-degree polynomial $y = a + bx + cx^2$ with $x = 1/w$ and $y$ can take on one of three values: $c_{loss}, c_{sharp}$ and $c_{max}$.

**Phase diagrams with depth**     Figure 20: shows the phase diagrams with depth for FCNs in NTP trained on the CIFAR-10 dataset. The phase diagrams look qualitatively similar compared to the $1/w$ phase diagrams.

## D   Relationship between various critical constants

Figure 21 illustrates the relationship between the early training critical constants for models and datasets. The experimental setup is the same as in Appendix C. Typically, we find that $c_{loss} \leq c_{sharp} \leq c_{sharp}$ holds true. However, there are some exceptions, which are observed at high values of $d/w$ (see 21 (d, e)), where the trends of the critical constants converge, and large fluctuations can cause deviations from the inequality.

## E   The effect of $d/w$ on the noise in dynamics

In this section, we demonstrate that for FCNs with $d/w \gtrsim 1/16$, the dynamics becomes noise-dominated. This aspect makes it challenging to distringuish the underlying deterministic dynamics from random fluctuations. To demonstrate this, we consider FCNs trained on CIFAR-10 using MSE and cross-entropy loss and use 4096 training examples for estimating sharpness.

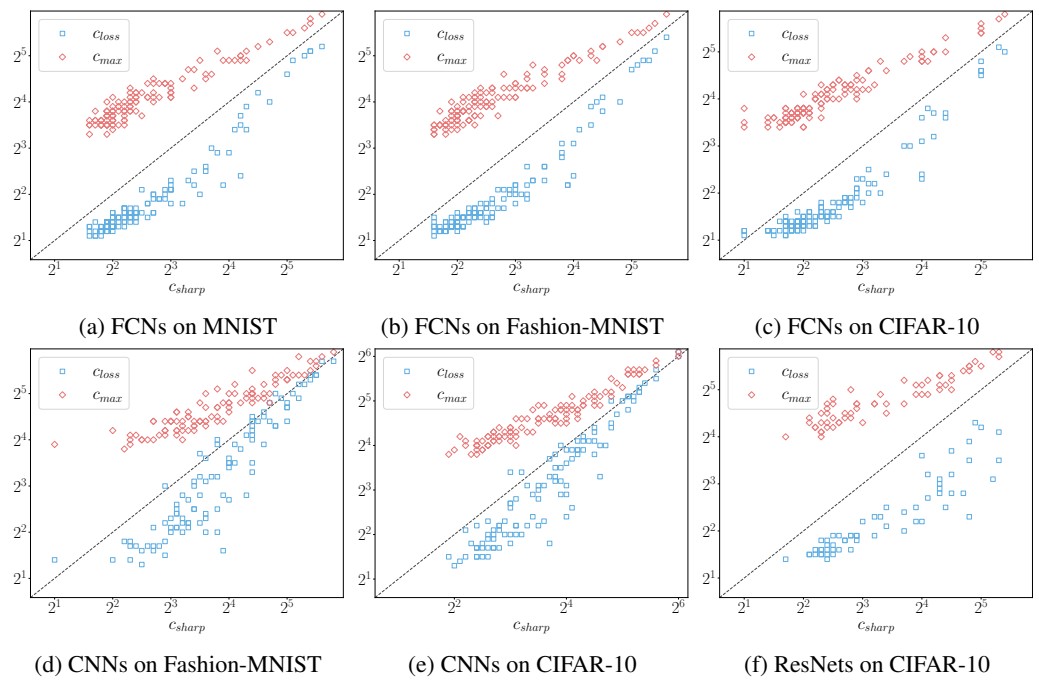

(a) FCNs on MNIST  (b) FCNs on Fashion-MNIST  (c) FCNs on CIFAR-10

(d) CNNs on Fashion-MNIST  (e) CNNs on CIFAR-10  (f) ResNets on CIFAR-10

Figure 21: The relationship between various critical constants for various models and datasets. Each data point corresponds to a model with random initialization. The dashed line denotes the values where $x = y$.

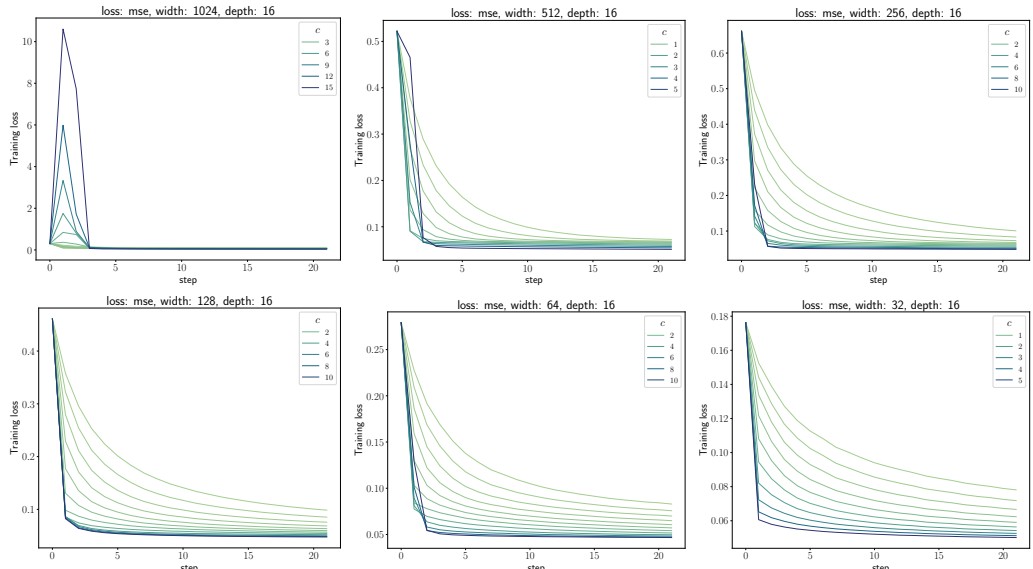

Figure 22: Training loss trajectories of ReLU FCNs with $d = 16$ trained on the CIFAR-10 dataset with MSE loss using SGD with learning rate $\eta = {}^c/\lambda_0^H$ and batch size $B = 512$.

Figures 22 and 23 show the training loss and sharpness of FCNs with $d = 16$ and varying widths, trained on CIFAR-10 using MSE loss. We observe that the sharpness dynamics becomes noisier for $w \lesssim 64$.

Figures 24 and 25 shows the training dynamics with loss switched to cross-entropy, while keeping the initialization and SGD batch sequence the same as in the MSE loss case. In comparison to MSE

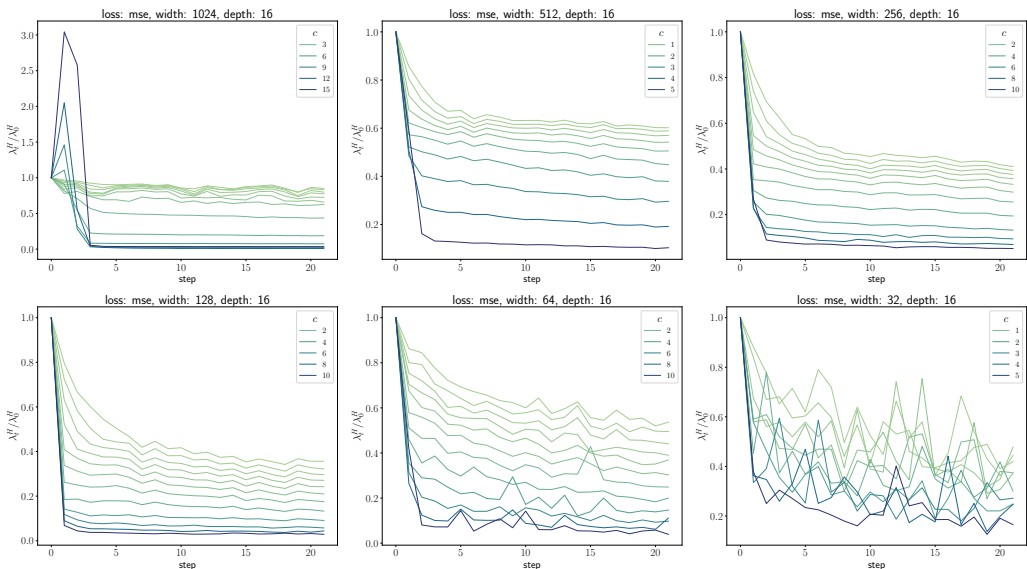

Figure 23: Sharpness trajectories of ReLU FCNs with $d = 16$ trained on the CIFAR-10 dataset with MSE loss using SGD with learning rate $\eta = c/\lambda_0^H$ and batch size $B = 512$.

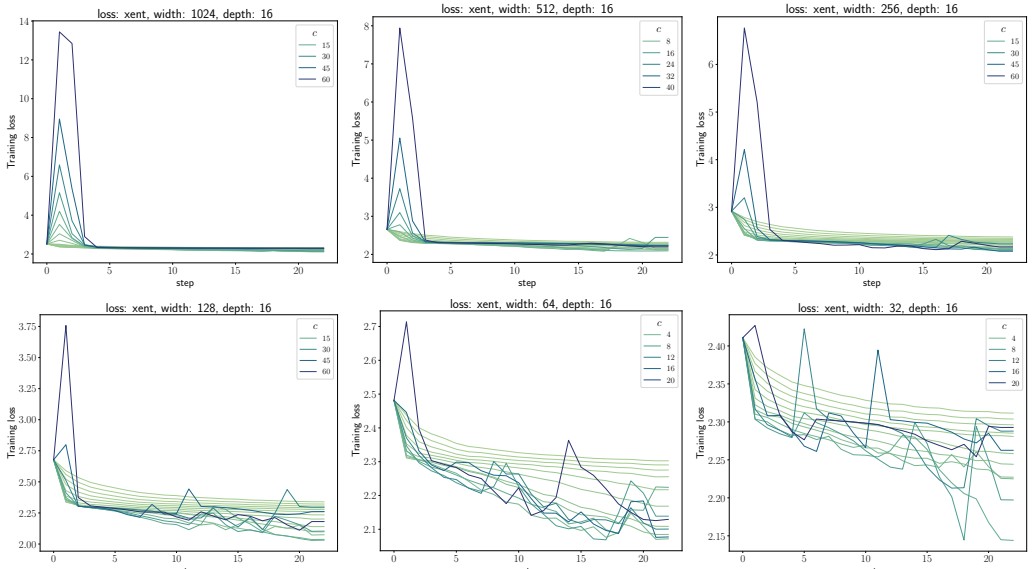

Figure 24: Training loss trajectories of ReLU FCNs with $d = 16$ trained on the CIFAR-10 dataset with cross-entropy loss using SGD with learning rate $\eta = c/\lambda_0^H$ and batch size $B = 512$.

loss, the training loss and sharpness dynamics show a higher level of noise, especially for $w \lesssim 256$. As a result, it becomes difficult to characterize the training dynamics for $d/w \gtrsim 1/16$.

## F  Crossentropy

In this section, we provide additional results for models trained with cross-entropy (xent) loss and compare them with MSE results. Broadly speaking, models trained with cross-entropy loss show similar characterstics to those trained with MSE loss, such as, (i) sharpness reduction during early training, (ii) an increase in critical constants $c_{loss}$, $c_{sharp}$ with $d$ and $1/w$, (iii) $c_{loss} \leq c_{sharp} \leq c_{max}$. However, the dynamics of models trained with cross-entropy loss is noisier compared to MSE as shown in the previous section, and characterizing these dynamics can be more complex. In the

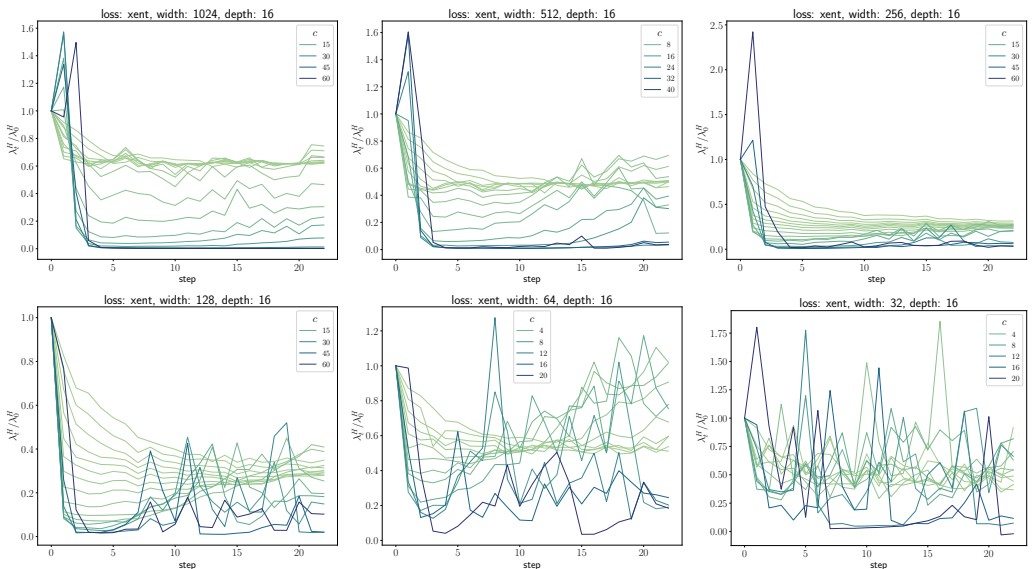

Figure 25: Sharpness trajectories of ReLU FCNs with $d = 16$ trained on the CIFAR-10 dataset with cross-entropy loss using SGD with learning rate $\eta = c/\lambda_0^H$ and batch size $B = 512$.

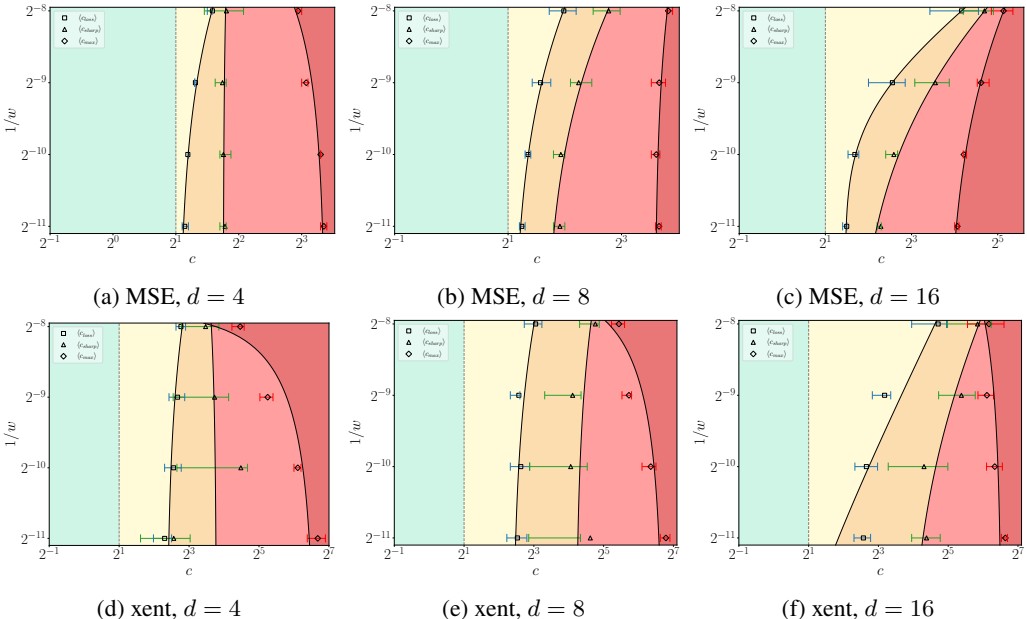

Figure 26: The phase diagrams of early training of FCNs trained on the CIFAR-10 dataset using (a, b, c) MSE and (d, e, f) cross-entropy loss. Each data point is an average over 10 initializations, and solid lines represent a smooth curve fitted to raw data points. The horizontal bars around the averaged data point indicates the region between 25% and 75% quantile. For cross-entropy phase diagrams, the $c = 2$ line is shown for reference only and does not relate to $c_{crit}$.

following experiments, we consider models trained on the CIFAR-10 dataset and used 4096 training examples to estimate sharpness.

### F.1 Phase diagrams

Figure 26 compares the phase diagrams of FCNs in SP trained on the CIFAR-10 dataset, using both MSE and cross-entropy loss. The estimated critical constants for cross-entropy loss are generally

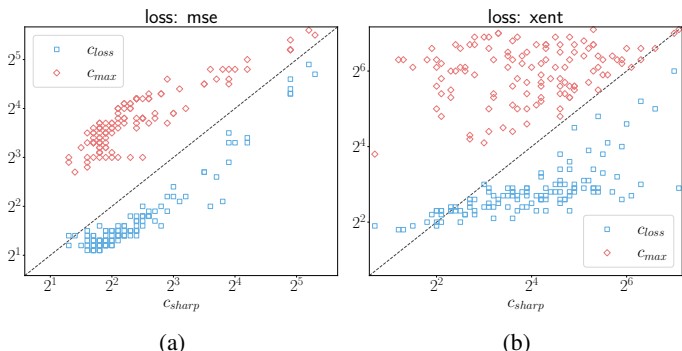

(a)                                   (b)

Figure 27: Comparison of the relationship between critical constants for FCNs in SP trained on CIFAR-10 using MSE and cross-entropy loss. Each data point corresponds to a randomly initialized model with depths and widths mentioned in Appendix A.

more noisy, as quantified by the confidence intervals. In comparison to phase diagrams of models trained with MSE loss, we observe a few notable differences. First, the loss starts to catapult at a value appreciably larger than $c = 2$ at large widths. Primarily, $4 \lesssim c_{loss} \lesssim 8$. Additionally, $c_{max}$ generally decreases with $1/w$. This decreasing trend becomes less sharp at large depths.

Despite these differences, the phase diagrams for both loss functions share various similarities. First, we observe sharpness reduces during early training for $c < c_{sharp}$ (see the first row of Figure 25). Next, we observe that the inequality $c_{loss} \leq c_{sharp} \leq c_{max}$ generally holds for both loss functions as demonstrated in Figure 27, barring some exceptions.

Figure 28 shows the phase diagrams for CNNs and ResNets trained on the CIFAR-10 dataset using cross-entropy loss. The observed critical constants are much noisier as quantified by the confidence intervals. Nevertheless, the phase diagram shows similar trends as mentioned above. For large $1/w$ models, we found that progressive sharpening begins after $5 - 10$ training steps. For these cases, we only use the first 5 steps to measure sharpness to avoid progressive sharpening. For CNNs, we observed that the dynamics becomes difficult to characterize for $w \lesssim 32$ and $d \gtrsim 10$, due to large fluctuations. Consequently, we've opted not to include these particular results.

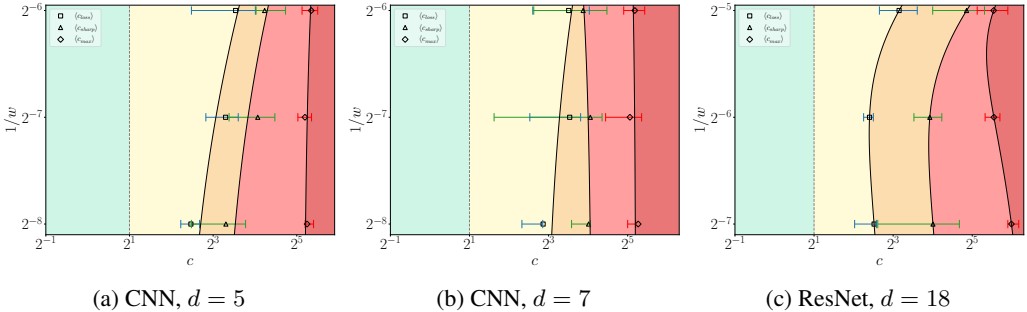

(a) CNN, $d = 5$            (b) CNN, $d = 7$            (c) ResNet, $d = 18$

Figure 28: Phase diagrams of (a, b) CNNs and (c) ResNets trained on the CIFAR-10 dataset with cross-entropy loss using SGD with $\eta = c/\lambda_0^H$ and $B = 512$.

## F.2   Intemediate saturation regime

Figure 29 shows the normalized sharpness measured at $c\tau = 100$ for FCNs trained on CIFAR-10 using cross-entropy loss. [7] Similar to MSE loss, we observe an abrupt drop in sharpness at large learning rates. However, this abrupt drop occurs at $2 \lesssim c_{crit} \lesssim 4$. The estimated sharpness is noisier (compare with Figure 38), which hinders a reliable estimation of $c_{crit}$. We speculate that we require a

---

[7]The time step $\tau = 100/c$ is in the middle of the intermediate saturation regime for most of the models. For further details on estimating sharpness, see Appendix I.1.

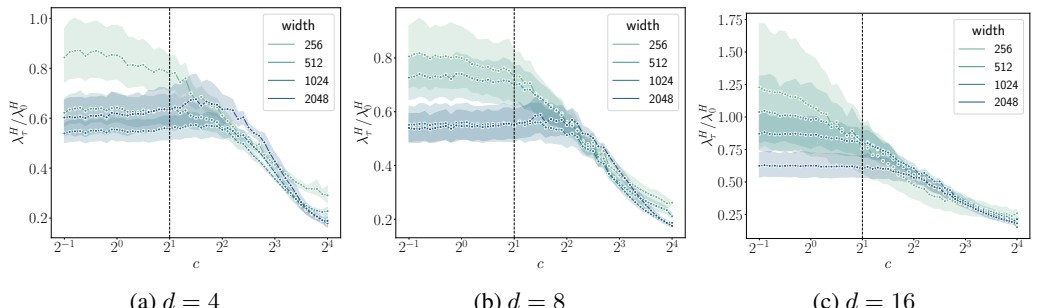

(a) $d = 4$             (b) $d = 8$             (c) $d = 16$

Figure 29: Sharpness measured at $c\tau = 100$ against the learning rate constant for FCNs trained on the CIFAR-10 dataset using cross-entropy loss, with varying depths and widths. Each curve is an average over ten initializations, where the shaded region depicts the standard deviation around the mean trend. The vertical dashed line shows $c = 2$ for reference.

large number of averages for a reliable estimation of $c_{crit}$ for cross-entropy loss. We leave the precise characterization of $c_{crit}$ for cross-entropy loss for future work.

## G    The effect of setting model output to zero at initialization

In this section, we demonstrate the effect of network output $f(x; \theta_t)$ at initialization on the early training dynamics. In particular, we set the network output to zero at initialization, $f(x; \theta_0) = 0$, by (1) 'centering' the network by its initial value $f_c(x; \theta_t) = f(x; \theta) - f(x; \theta_0)$ or (2) setting the last layer weights to zero at initialization. We show that both (1) and (2) remove the opening of the sharpness reduction phase with $1/w$. Resultantly, the average onset of loss catapult occurs at $c_{loss} \approx 2$, independent of depth and width.

Throughout this section, we use 'vanilla' networks to refer to networks initialized in the standard way. For simplicity, we train FCNs using full batch gradient descent with MSE loss using a subset consisting of 4096 examples of the CIFAR-10 dataset.

### G.1    The effect of centering networks

Given a network function $f(x; \theta_t)$, we define the centered network $f_c(x; \theta_t)$ as

$$f_c(x; \theta_t) = f(x; \theta_t) - f(x; \theta_0), \tag{45}$$

where $f(x; \theta_0)$ is the network output at intialization. By construction, the network output is zero at initialization. It is noteworthy that centering a network is an unusual way of training deep networks as it doubles the cost of training because of two forward passes.

Figure 30 compares the training loss and sharpness dynamics of vanilla networks and centered networks. Unlike vanilla networks, we do not observe a decrease in sharpness for $c < c_{loss}$ during early training. Rather, we observe a slight increase in sharpness. To distinguish this slight increase from sharpness catapult, we introduce a threshold $\epsilon$, comparing normalized sharpness $\lambda_t^H / \lambda_0^H$ with $1 + \epsilon$, to define a sharpness catapult.[8] As demonstrated in Appendix G.3, the $uv$ model trained on a single training example $(x, y)$ with $y \neq 0$ sheds lights on this initial increase in sharpness.

Interestingly, irrespective of depth and width, we observe that loss catapults at $c_{loss} \approx 2$, as demonstrated in the phase diagrams in Figure 31(a, b, c). These findings suggest a strong correlation between a large network output at initialization $\|f(x; \theta_0)\|$ and the opening of the sharpness reduction phase discussed in Section 2.

### G.2    The effect of setting the last layer to zero

An alternative way to train networks with $f(x; \theta_0) = 0$ is by setting the last layer to zero at initialization. The principle of criticality at initialization [55, 58, 68] does not put any constraints on

---

[8]In experiments, we set $\epsilon = 0.05$. We use the same threshold for zero-init networks.

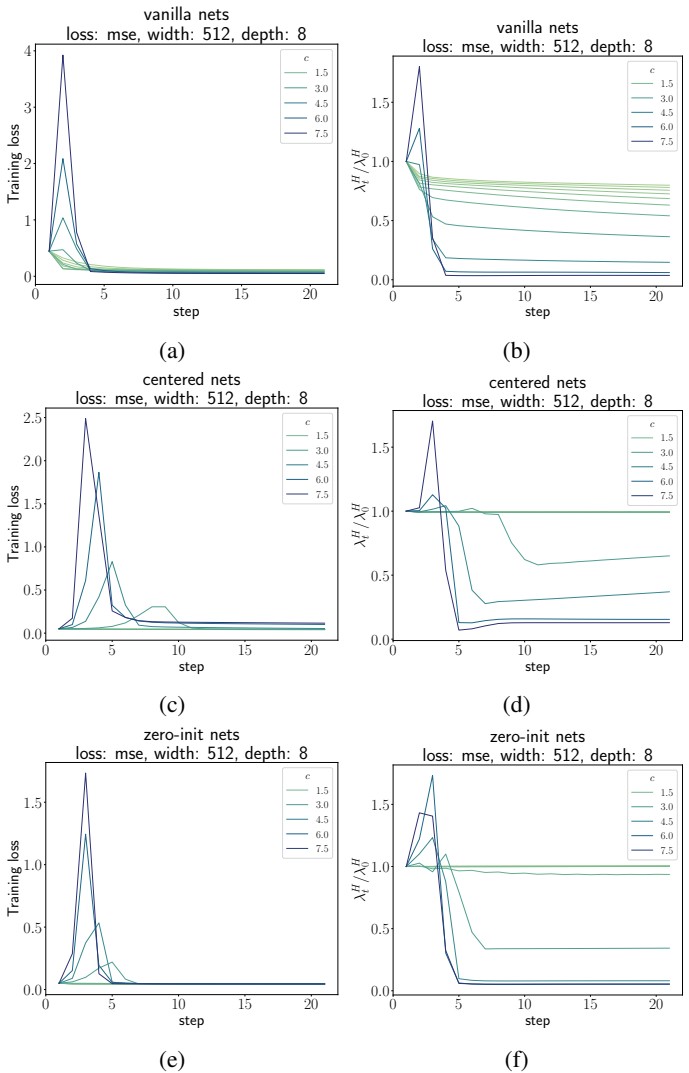

Figure 30: Comparison of the early training dynamics of (a, b) vanilla, (c, d) centered, and (e, f) zero-initialized FCNs (with depth $= 8$ and width $= 512$), trained on the CIFAR-10 dataset with MSE loss using gradient descent for 20 steps.

the last layer weights. Hence, setting the last layer to zero does not affect signal/gradient propagation at initialization. Yet, setting the last layer to zero results in initialization in a flat curvature region at initialization, resulting in access to larger learning rates. We refer to these networks as 'zero-init' networks.

Figure 30 compares the training dynamics of zero-init networks with vanilla and centered networks. We observe that the dynamics is quite similar to the centered networks: (i) sharpness does not reduce for small learning rates and (ii) loss catapults $c_{loss} \approx 2$, irrespective of depth and width. Figure 31(d, e, f) show the phase diagrams of networks with zero-initialized networks. Like centered networks, the critical constants do not scale with depth and width. Again, suggesting that a large network output at initialization $\|f(x; \theta_0)\|$ is related to the opening of the sharpness reduction phase in the early training results shown in Section 2.

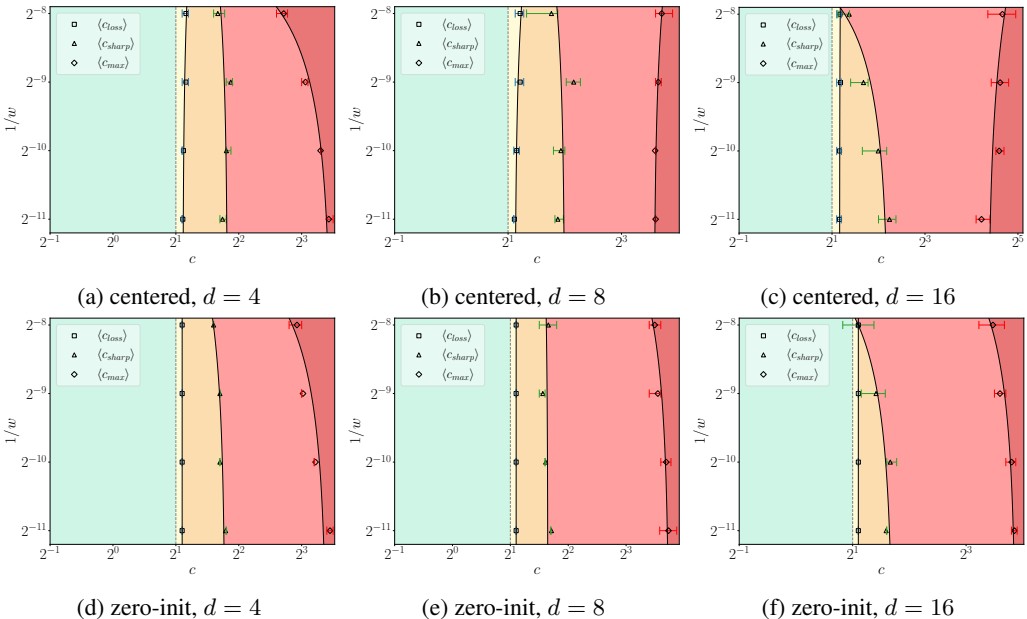

(a) centered, $d = 4$    (b) centered, $d = 8$    (c) centered, $d = 16$

(d) zero-init, $d = 4$    (e) zero-init, $d = 8$    (f) zero-init, $d = 16$

Figure 31: The phase diagrams of early training dynamics of (a, b, c) centered and (d, e, f) zero-init networks trained on CIFAR-10 using MSE using gradient descent. Each data point is an average over 10 initializations. The horizontal bars around the average data point indicate the region between 25% and 75% quantile.

### G.3  Insights from $uv$ model trained on $(x, y)$

In this section, we gain insights into the effect of setting network output to zero at initialization using $uv$ model trained on an example $(x, y)$. In particular, we show that loss catapults at $k_{loss} = 2$ and sharpness increases during early training.

Consider the $uv$ model trained on a single training example $(x, y)$ with $y \neq 0$ [9]

$$f(x) = \frac{1}{\sqrt{w}} \sum_i^w u_i v_i \, x.$$

This simplifies the loss function to

$$\mathcal{L} = \frac{1}{2} \left( f(x) - y \right)^2 = \frac{1}{2} \Delta f^2, \tag{46}$$

where $\Delta f$ is the residual. The trace of the Hessian $\mathrm{tr}(H)$ is

$$\mathrm{tr}(H) = \frac{x^2}{w} \left( \|v\|^2 + \|u\|^2 \right). \tag{47}$$

The Frobeinus norm can be written in terms of the trace and the network output

$$\|H\|_F^2 = \lambda^2 + 2x^2 \Delta f^2 \left( 1 + \frac{2f}{w \Delta f} \right). \tag{48}$$

The function and residual updates are given by

---

[9]Note that for $y = 0$, the network is already at a minimum for $f_0 = 0$.

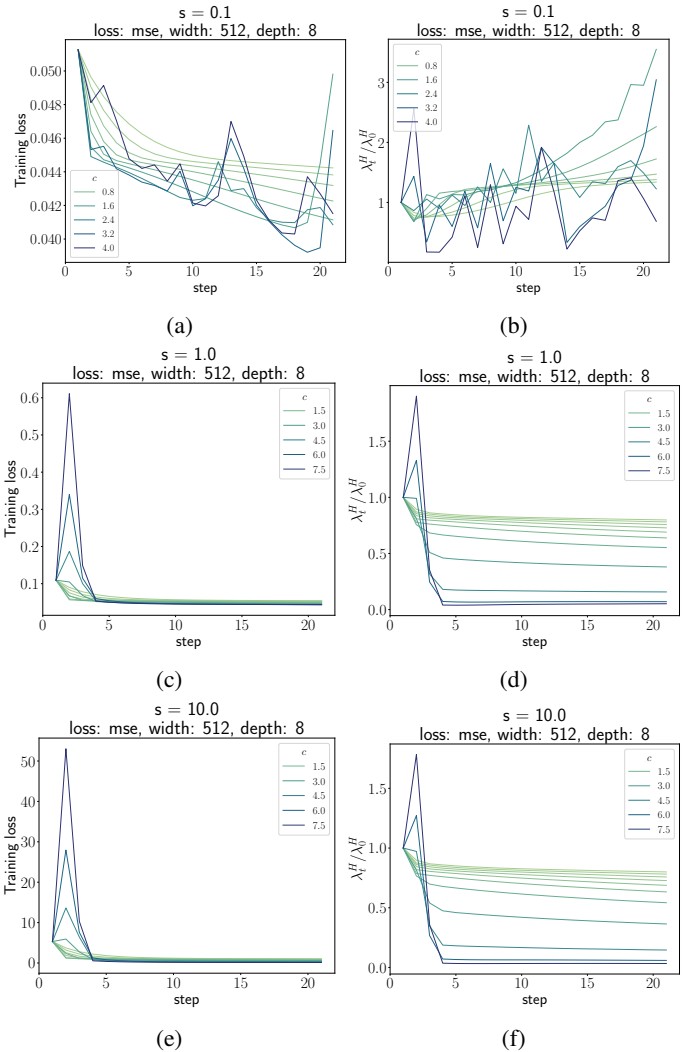

Figure 32: The early training dynamics of FCNs with a fixed output scale trained on the CIFAR-10 dataset with MSE loss using gradient descent.

$$f_{t+1} = f_t - \eta \operatorname{tr}(H_t) + \frac{\eta^2 x^2}{w} f_t \Delta f_t^2 \tag{49}$$

$$\Delta f_{t+1} = \Delta f_t \left(1 - \eta \operatorname{tr}(H_t) + \frac{\eta^2 x^2}{w} f_t \Delta f_t\right). \tag{50}$$

Similarly, we can obtain the trace update equations

$$\operatorname{tr}(H_{t+1}) = \operatorname{tr}(H_t) + \frac{\eta \Delta f_t^2 x^2}{w} \left(\eta \operatorname{tr}(H_t) - 4\frac{f_t}{\Delta f_t}\right). \tag{51}$$

Let us analyze them for the networks with zero output at initialization. The loss at the first step increases if

$$\left\langle \frac{L_1}{L_0} \right\rangle = \left\langle \left(1 - \eta \operatorname{tr}(H_0) + \frac{\eta^2 x^2}{n} f_0 \Delta f_0\right)^2 \right\rangle > 1 \tag{52}$$

$$\tag{53}$$

Setting $f_0 = 0$ and scaling the learning rate as $\eta = k/\operatorname{tr}(H_0)$, we see that the loss increases at the first step if $k > 2$.

$$\left\langle \frac{L_1}{L_0} \right\rangle = \left\langle (1 - k)^2 \right\rangle > 1 \tag{54}$$

Next, we analyze the change in trace during the first training step. Setting $f_0 = 0$, we observe that the trace increases for all learning rates

$$\operatorname{tr}(H_1) = \operatorname{tr}(H_0) + \frac{\eta^2 x^2}{w} \Delta f_0^2 \operatorname{tr}(H_0), \tag{55}$$

modulated by the learning rate and width. Finally, we analyze the change in Frobenius norm in the first training step at $k = k_{loss}$, which implies $\Delta f_1^2 = \Delta f_0^2$,

$$\left\langle \Delta \|H_1\|^2 \right\rangle = \left\langle \operatorname{tr}(H_1)^2 - \operatorname{tr}(H_0)^2 + 2x^2 \left(\Delta f_1^2 - \Delta f_0^2\right) \right\rangle. \tag{56}$$

As $\operatorname{tr}(H)$ increases in the first training step, $\|H\|_F$ also increases in the first training step.

## H   The effect of output scale on the training dynamics

Given a neural network function $f(x)$ with depth $d$ and width $w$, we define the scaled network as $f_s(x) = \alpha f(x)$, where $\alpha$ is referred to as the output scale. In this section, we empirically study the impact of the output scale on the early training dynamics. In particular, we show that a large (resp. small) value of $\|f(x; \theta_0)\|$ relative to the one-hot encodings of the labels causes the sharpness to decrease (resp. increase) during early training. Interestingly, we still observe an increase in $\langle c_{loss} \rangle$ with $d$ and $1/w$, unlike the case of initializing network output to zero, highlighting the unique impact of output scale on the dynamics. For simplicity, we train FCNs using gradient descent with MSE loss using a subset consisting of 4096 examples of the CIFAR-10 dataset, as in the previous section.

### H.1   The effect of fixed output scale at initialization

In this section, we study the training dynamics of models trained with a fixed output scale at initialization. Given a network output function $f(\theta)$, we define the 'scaled network' as

$$f_s(\theta) = \frac{s f(\theta)}{\|f(\theta_0)\|}, \tag{57}$$

where $s$ is a scalar, fixed throughout training. By construction, the network output norm $\|f_s(\theta_0)\|$ equals $s$. For standard initialization, $s = \|f(\theta_0)\| = \mathcal{O}(\sqrt{k})$, where $k$ are the number of classes.

Figure 32 shows the training dynamics of FCNs for three different values of the output scale $s$. The training dynamics of networks with $s = 1.0$ and $s = 10.0$ share qualitative similarities. In contrast, networks initialized with a smaller output scale ($s = 0.1$) exhibit distinctly different dynamics. In particular, we observe that for large output scales ($s \gtrsim 0.5$) sharpness decreases during early training, while sharpness increases for small output scales [10]. Furthermore, the training dynamics tends to

---

[10]We empirically observed that sharpness reduces for output scales as small as $s \sim 0.5$, which is relatively small compared to $\sqrt{k}$.

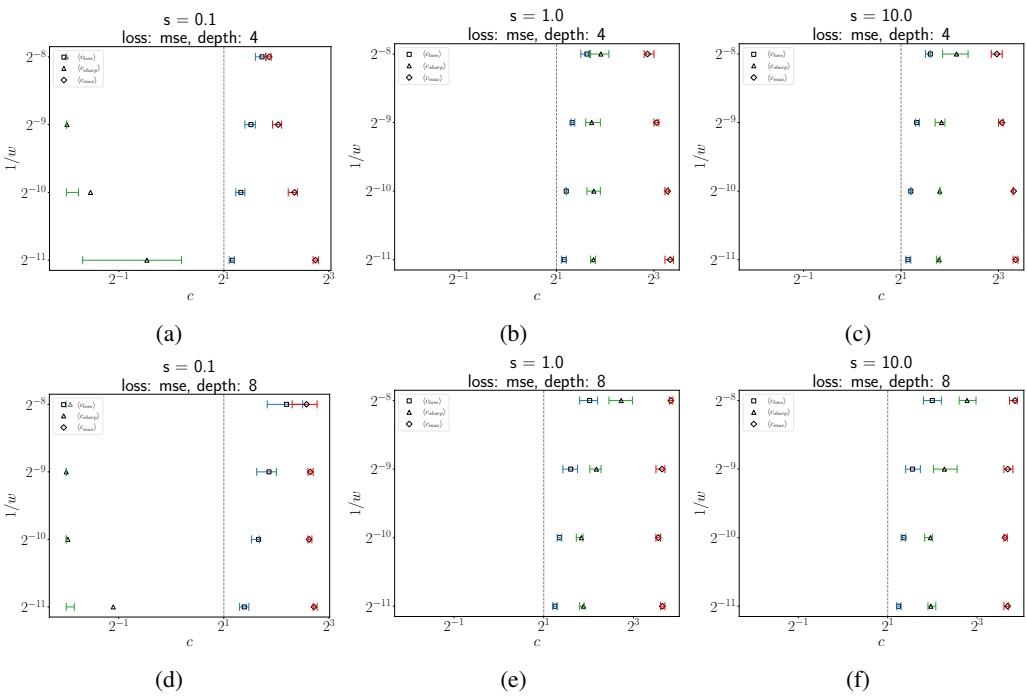

(a)          (b)          (c)

(d)          (e)          (f)

Figure 33: The phase diagrams of early training dynamics for ReLU FCNs with fixed output scale trained on a subset of the CIFAR-10 dataset using MSE loss using gradient descent. Each data point is an average over 10 initializations. The horizontal bars around the average data point indicate the region between 25% and 75% quantile.

be noisier at small output scales, making it difficult to characterize catapult dynamics amidst these fluctuations. In summary, the training dynamics of networks with small output scale deviate from the training dynamics discussed in the main text, particularly as the sharpness quickly increases during early training.

Figure 33 shows the trends of various critical constants with width for FCNs for three different values of $s$. Similar to vanilla networks, we observe that $c_{loss}$ increases with $d$ and $1/w$. In comparison, sharpness decreases (increases) for large (small) values of $s$. These experiments suggest that the output scale primarily influences the increase/decrease in sharpness during early training and does not affect the scaling of $c_{loss}$ with depth and width.

Note that we do not generate phase diagrams for these experiments as the training dynamics of networks with small output scales at initialization deviate from the training dynamics disucssed in the main text.

## H.2   Scaling the output scale with width

In this section, we study the training dynamics of models with an output scale scaled with width as $\alpha = w^{-\sigma}$, which is commonly used in the literature [19, 6, 4]. We consider three distinct $\sigma$ values $\{-0.5, 0.0, 0.5\}$, where $\sigma = -0.5$ represents the lazy regime, $\sigma = 0.5$ corresponds to feature learning (rich) regime and $\sigma = 0.0$ correponds to standard (vanilla) initialization.

Figure 34 shows the training loss and sharpness trajectories of FCNs trained on for different $\sigma$ values. We observe that the training trajectories in the lazy regime look identical to standard initialization. In comparison, the training trajectories in the feature learning regime is distinctly different. We observe that in the standard and lazy regimes, sharpness decreases during early training, whereas sharpness tends to increase in the feature learning regime and eventually oscillates around the edge of stability regime. Moreover, we observe that sharpness can catapult before the training loss in the feature learning regime (compare catapult peaks in 34(e, f)). These results are in parallel to the fixed output scale networks studied in the pervious section.

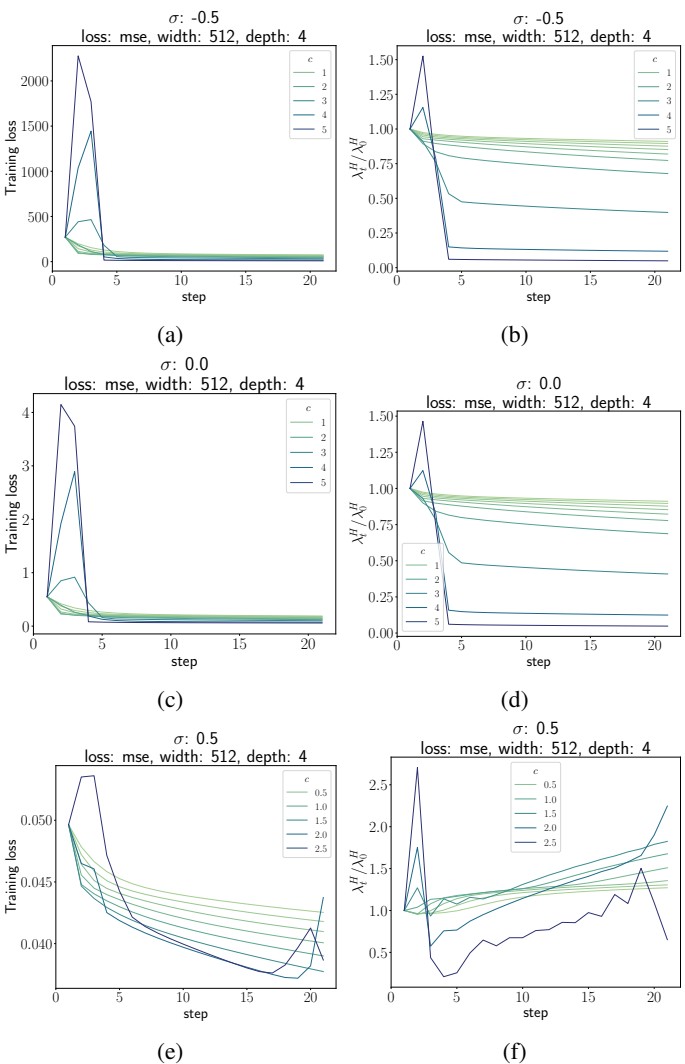

Figure 34: The early training dynamics of FCNs with output scale $\alpha = w^{-\sigma}$ trained on the CIFAR-10 dataset with MSE loss using gradient descent.

Figure 35 summarizes the early training dynamics of FCNs with different $\sigma$ values. We observe similar results as in the previous section. The output scale affects the initial increase/decrease of sharpness but does not affect the scaling trend of $c_{loss}$ with depth and width. Moreover, we observe a systematic pattern of $c_{max}$ scaling with width. In the lazy regime, we observe that $c_{max}$ increases with $1/w$, while $c_{max}$ decreases with $1/w$ in the feature learning regime.

# I    Sharpness curves in the intermediate saturation regime

This section shows additional results for Section 3 for MSE loss. Cross-entropy results are shown in Appendix F. Figures 36 to 40 show the normalized sharpness curves for different depths and widths.

## I.1    Estimating the sharpness

This paragraph describes the procedure for measuring the sharpness to study the effect of the learning rate, depth, and width in the intermediate saturation regime. We measure the sharpness $\lambda_\tau^H$ at a time $\tau$ in the middle of the intermediate saturation regime. We choose $\tau$ so that $c\tau \approx 200$, for learning rates $c = 2^x$, where $x \in [-1.0, 4.0]$ in steps of $0.1$. The value 200 is chosen such that $\tau$ is in the middle of the intermediate saturation regime. Next, we measure sharpness over a range of steps

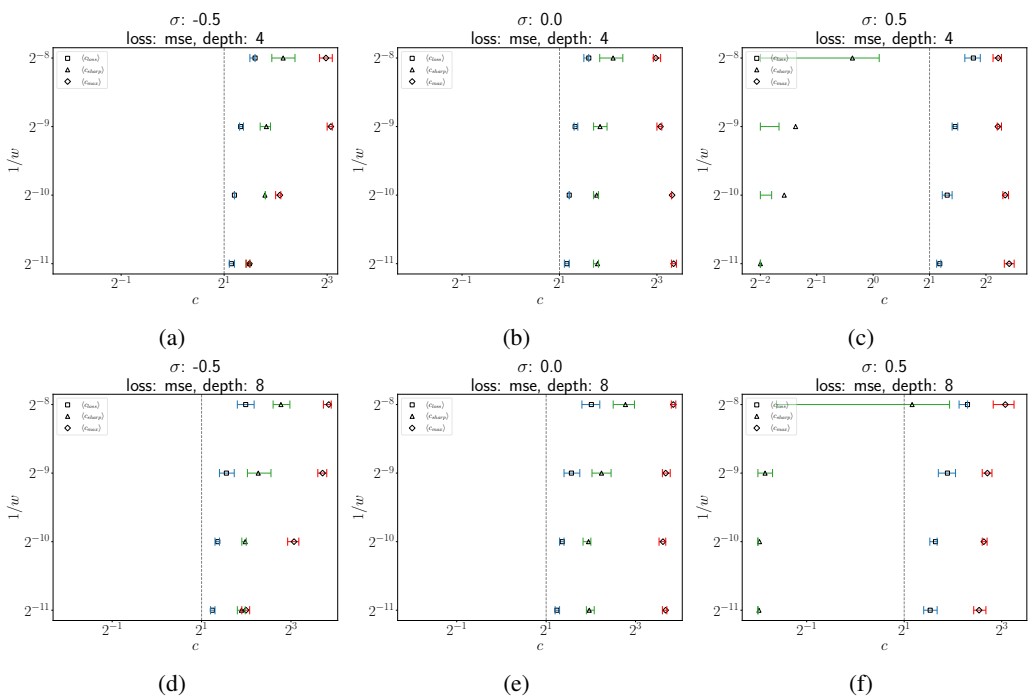

Figure 35: The phase diagrams of early training dynamics for ReLU FCNs with varying depths and output scale.

$t \in [\tau - 5, \tau + 5]$ and average over $t$ to reduce fluctuations. We repeat this process for various initializations and obtain the average sharpness.

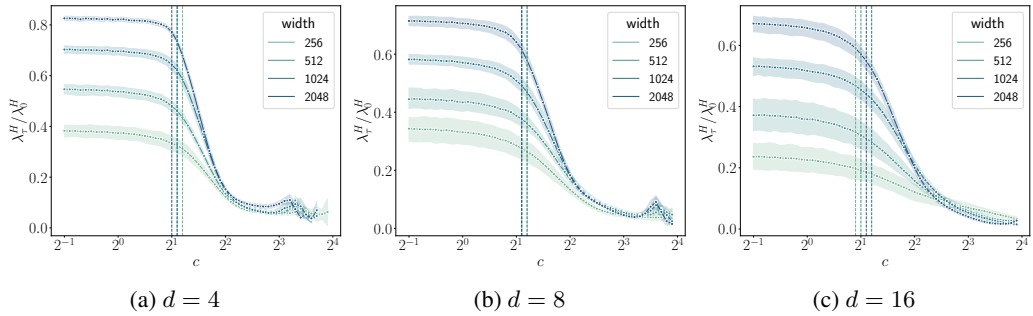

(a) $d = 4$        (b) $d = 8$        (c) $d = 16$

Figure 36: Sharpness measured at $c\tau = 200$ against the learning rate constant for FCNs trained on the MNIST dataset, with varying depths and widths. Each curve is an average over ten initializations, where the shaded region depicts the standard deviation around the mean trend. The vertical lines denote $c_{crit}$ estimated using the maximum of $\chi'_\tau$.

## I.2    Estimating the critical constant $c_{crit}$

This subsection explains how to estimate $c_{crit}$ from sharpness measured at time $\tau$. First, we normalize the sharpness with its initial value, and then average over random initializations. Next, we estimate the critical point $c_{crit}$ using the second derivative of the order parameter curve. Even if the obtained averaged normalized sharpness curve is somewhat smooth, the second derivative may become extremely noisy as minor fluctuations amplify on taking derivatives. This can cause difficulties in obtaining $c_{crit}$. We resolve this issue by estimating the smooth derivatives of the averaged order parameter with the Savitzky–Golay filter [59] using its scipy implementation [63]. The estimated $c_{crit}$ is shown by vertical lines in the sharpness curves in Figures 36 to 40.

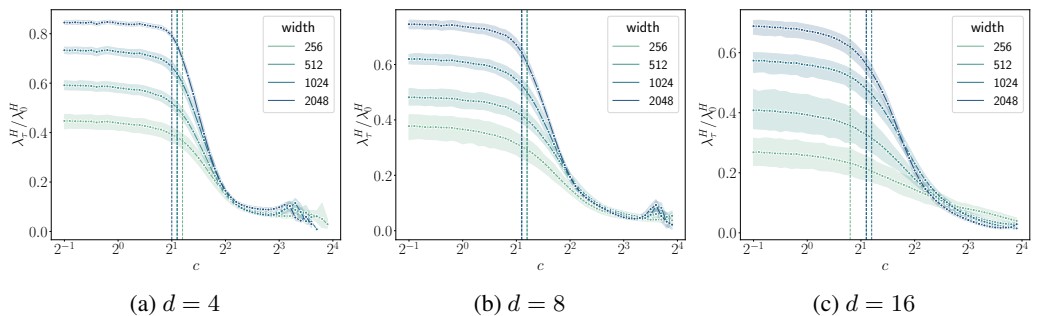

(a) $d = 4$       (b) $d = 8$       (c) $d = 16$

Figure 37: Sharpness measured at $c\tau = 200$ against the learning rate constant for FCNs trained on the Fashion-MNIST dataset, with varying depths and widths. Each curve is an average over ten initializations, where the shaded region depicts the standard deviation around the mean trend. The vertical lines denote $c_{crit}$ estimated using the maximum of $\chi'_{\tau}$.

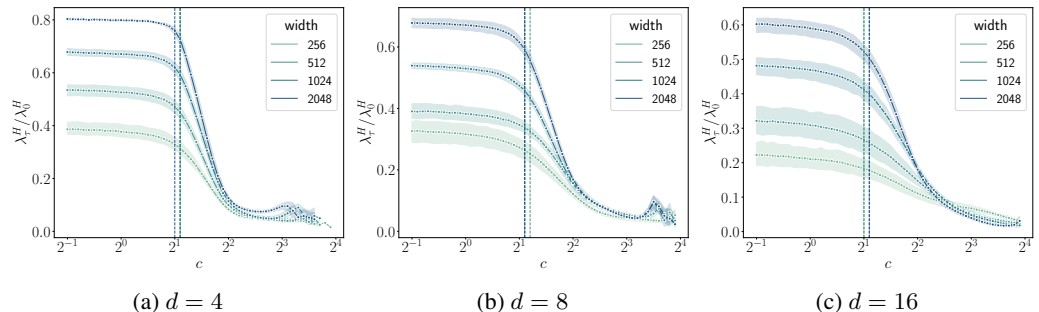

(a) $d = 4$       (b) $d = 8$       (c) $d = 16$

Figure 38: Sharpness measured at $c\tau = 200$ against the learning rate constant for FCNs trained on the CIFAR-10 dataset, with varying depths and widths. Each curve is an average over ten initializations, where the shaded region depicts the standard deviation around the mean trend. The vertical lines denote $c_{crit}$ estimated using the maximum of $\chi'_{\tau}$.

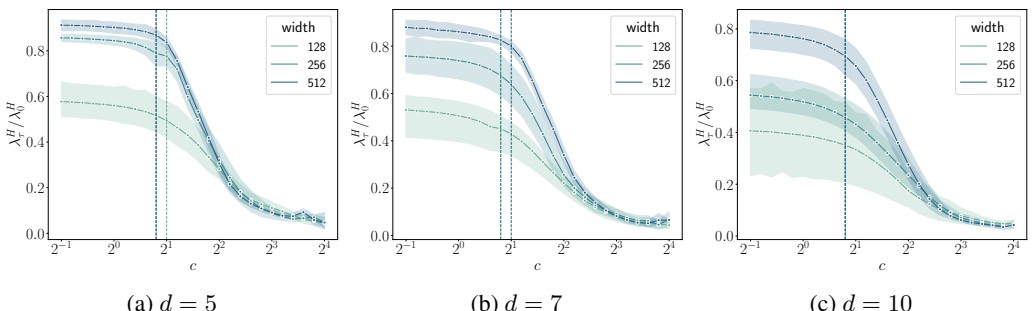

(a) $d = 5$       (b) $d = 7$       (c) $d = 10$

Figure 39: Sharpness measured at $c\tau = 200$ against the learning rate constant for Myrtle-CNNs trained on the CIFAR-10 dataset, with varying depths and widths. Each curve is an average of over ten initializations, where the shaded region depicts the standard deviation around the mean trend. The vertical lines denote $c_{crit}$ estimated using the maximum of $\chi'_{\tau}$.

## J The effect of batch size on the reported results

### J.1 The early transient regime

Figure 41 shows the phase diagrams of early training dynamics of FCNs with $d = 4$ trained on the CIFAR-10 dataset using two different batch sizes. The phase diagram obtained is consistent with the findings presented in Section 2, except for one key difference. Specifically, we observe that when $d/w$ is small and small batch sizes are used for training, sharpness may increase from initialization

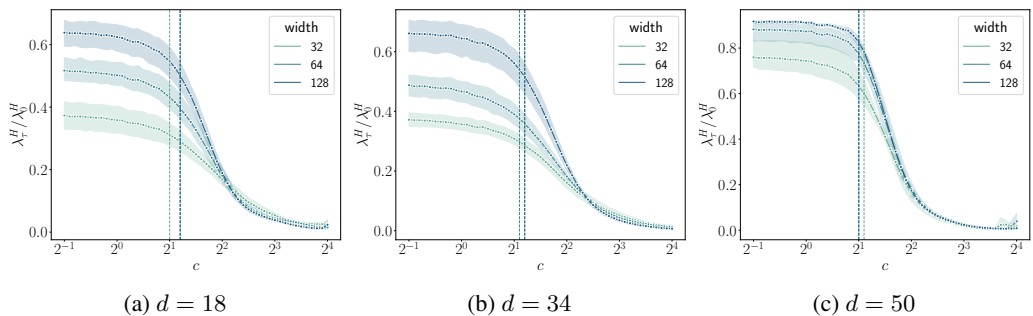

(a) $d = 18$  (b) $d = 34$  (c) $d = 50$

Figure 40: Sharpness measured at $c\tau = 200$ against the learning rate constant for ResNets trained on the CIFAR-10 dataset, with varying depths and widths. Each curve is an average of over ten initializations, where the shaded region depicts the standard deviation around the mean trend. The vertical lines denote $c_{crit}$ estimated using the maximum of $\chi'_\tau$.

at relatively smaller values of $c$. This is reflected in Fig. 41 by $\langle c_{sharp} \rangle$ moving to the left as $B$ is reduced from 512 to 128. However, this initial increase in sharpness is small compared to the sharpness catapult observed at larger batch sizes. We found that this increase at small batch sizes is due to fluctuations in gradient estimation that can cause sharpness to increase above its initial value by chance.

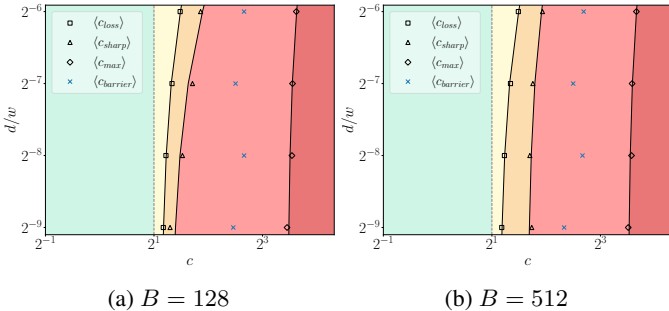

(a) $B = 128$  (b) $B = 512$

Figure 41: The phase diagram of early training for FCNs with $d = 4$ trained on the CIFAR-10 dataset with MSE loss using SGD with different batch sizes.

## J.2 The intermediate saturation regime

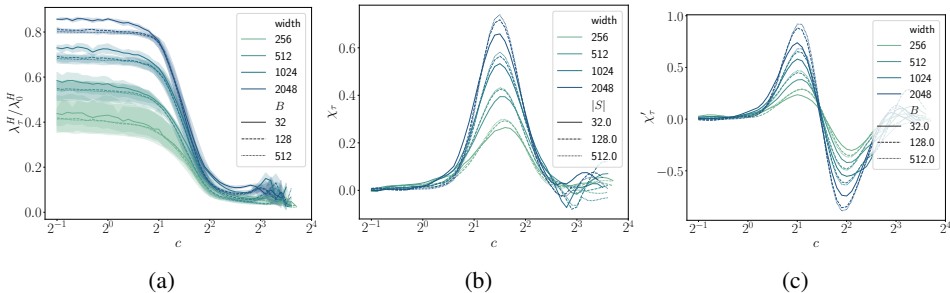

(a)  (b)  (c)

Figure 42: (a) Normalized sharpness measured at $c\tau = 200$ against the learning rate constant for FCNs with $d = 4$ trained on the CIFAR-10 dataset, with varying widths. Each data point is an average over 10 initializations, where the shaded region depicts the standard deviation around the mean trend. (b, c) Smooth estimations of the first two derivatives, $\chi_\tau$ and $\chi'_\tau$, of the averaged normalized sharpness wrt the learning rate constant.

Figure 42 shows the normalized sharpness, measured at $c\tau = 200$, and its derivatives for various widths and batch sizes. The results are consistent with those in Section 3, with a lowering in the peak

heights of the derivatives $\chi$ and $\chi'$ at small batch sizes. The lowering of the peak heights means the full width at half maximum increases, which implies a broadening of the transition around $c_{crit}$ at smaller batch sizes.

## K    The effect of bias on the reported results

In this section, we show that FCNs with bias show similar results as presented in the main text. We considered FCNs in SP initialized with He initialization [29].

Figure 43 shows the phase diagrams of early training for FCNs with bias trained on the CIFAR-10 dataset. We observe a similar phase diagram compared to the no-bias case (compare with Figure 26).

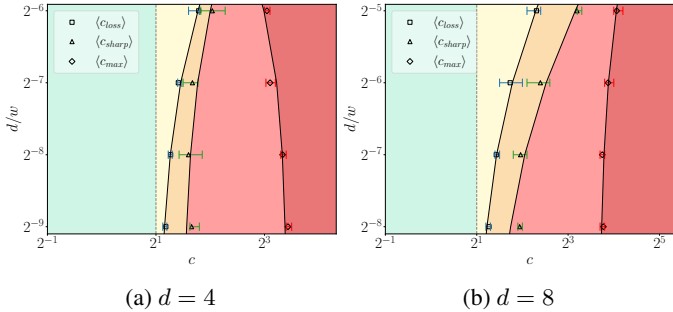

(a) $d = 4$                    (b) $d = 8$

Figure 43: The phase diagram of early training for FCNs with bias trained on the CIFAR-10 dataset with MSE loss using SGD with different depths.

