# OpenReview forum: "Phase diagram of early training dynamics in deep neural networks: effect of the learning rate, depth, and width"
_NeurIPS.cc/2023/Conference — NeurIPS 2023 poster_

### Official Review · Reviewer_bhX5 · 2023-06-16

**Soundness:** 4 excellent
**Presentation:** 4 excellent
**Contribution:** 3 good
**Rating:** 7
**Confidence:** 4

**Summary:**

The paper studies the effect of depth, width and learning rate on the early training dynamics of DNNs. The authors first describe four phases throughout, and then focus on the first two phases.

In the first phase, they identify three ranges of learning rates, according to how they affect the evolution of the training loss and the sharpness of the loss in the first few training steps:
- for small learning rates both loss and sharpness decrease monotonically. Surprisingly, this regime extends to learning rates that are larger than the traditional threshold of $2 / \lambda_{max}$ which guarantees a close match of GD to GF.
- For larger learning rates, the loss is non-monotone but the sharpness still decreases.
- For yet larger learning rates, both the loss and sharpness are non-monotone.
Any larger learning rates lead to divergence.
In general larger depths $d$ and smaller widths $w$ shifts these regime to larger values of the learning rate.

In the second phase, the sharpness settles to some value. For small learning rates $\eta$, this value is roughly constant w.r.t. $\eta$ (but which depends on the width), but for large $\eta$ the sharpness decreases and settles to a value which seems to be independent of the width.

The simple model of a shallow linear network with one datapoint and one dimensional inputs and outputs is studied, and the qualitative features described above are recovered.

**Strengths:**

Understanding the different regimes of training as a function of width, depth and learning rates is an important problem, and any work that studies these questions either mathematically or empirically is welcome. The analysis and discussions are clear and insightful. A range of different architectures are tested. The regimes proposed and studied are to my knowledge new.

**Weaknesses:**

The authors define a large number of regimes acoording to some rather subtle changes in behavior sometimes, and it is not always clear whether these regimes are also correlated with other quantities that we actually care about (e.g. generalization/feature learning). The training phase of neural networks features many strange/non-monotonic behavior and it is not clear that all of them have to be identified, since they might have a rather small impact.


**Questions:**

It is shortly mentioned that setting the outputs of the network at initialization to zero changes the ranges of learning rates corresponding to the regimes of phase 1. In the appendix, it appears that the transitions between ranges become almost independent of the width in this case, and the smallest range (where both loss and sharpness decrease) stops much closer to the expected value of $2 / \lambda_{max}$. This suggests that the dependence of these values on the width $w$ which is quite thoroughly studied in the main paper, might only be related to the variance size of the outputs at initialization, and not some deeper property of the network. Ideally I think that the plots of the appendix describing this phenomenon should be put in the main, as it greatly impacted my personal interpretation of the results of the paper.

Did the authors also check the behavior of the output function norm in the first few timesteps? I was wondering whether the non-monotonic behavior at the begining of training might be related to the function norm having a similar non-monotonic behavior (scaling up then down, possibly to some value close to zero), since the function norm has an obvious impact on both the loss and sharpness of the loss.


**Limitations:**

The limitations of the results presented are discussed well.

---

> ### Author Rebuttal · Authors · 2023-08-07
>
> We thank the reviewer for their encouraging comments. Below are our responses to the comments:
>
> > The authors define a large number of regimes acoording to some rather subtle changes in behavior sometimes, and it is not always clear whether these regimes are also correlated with other quantities that we actually care about (e.g. generalization/feature learning). The training phase of neural networks features many strange/non-monotonic behavior and it is not clear that all of them have to be identified, since they might have a rather small impact.
>
> We believe that analyzing the generalization and feature learning properties through the lens of our phase diagram could lead to a more comprehensive understanding. Various studies, including [46, 66] and a recent paper (https://arxiv.org/abs/2306.04815), claim that large learning rates lead to better performance.
> In particular, [46] conjectures that optimal performance in wide networks is obtained for learning rates between $c_{loss} = 2$ and $c_{max}$. We believe that our finer characterization of the phase diagram, particularly at finite depth and width, could help in narrowing down the range of learning rates with optimal performance.
>
> From a broader perspective, we believe that any systematic behavior in optimization dynamics that occurs repeatedly throughout architectures and datasets is worth documenting and trying to understand. While we sympathize with the referee's comment that it is unclear what will be useful for quantities we care about, we do think that systematic studies that shed light on the nature of the loss landscape and the behavior of nonconvex optimization are worthwhile as it is difficult to predict which discoveries will eventually be practically useful.
>
>
> > It is shortly mentioned that setting the outputs of the network at initialization to zero changes the ranges of learning rates corresponding to the regimes of phase 1. In the appendix, it appears that the transitions between ranges become almost independent of the width in this case, and the smallest range (where both loss and sharpness decrease) stops much closer to the expected value of $2 / \lambda_{max}$. This suggests that the dependence of these values on the width $w$ which is quite thoroughly studied in the main paper, might only be related to the variance size of the outputs at initialization, and not some deeper property of the network. Ideally I think that the plots of the appendix describing this phenomenon should be put in the main, as it greatly impacted my personal interpretation of the results of the paper.
>
> We understand that this aspect may deserve more attention in the main text. Accordingly, we can move the relevant phase diagrams (for example, Figure 30) from the appendix to the main part of the paper to highlight this observation.
>
> > Did the authors also check the behavior of the output function norm in the first few timesteps? I was wondering whether the non-monotonic behavior at the begining of training might be related to the function norm having a similar non-monotonic behavior (scaling up then down, possibly to some value close to zero), since the function norm has an obvious impact on both the loss and sharpness of the loss.
>
> Based on the reviewer's feedback, we performed a preliminary analysis of the output function norm during early training in FCNs. For models trained with MSE loss, we found that the critical learning rate constant $c_{output}$ at which the output norm increases from initialization aligns with $c_{loss}$ (see table below). This is expected because the output norm is closely related to the loss. However, for cross-entropy loss, we found that the critical constant for the output norm is smaller than $c_{loss}$ as shown in the table below, specifically, $c_{output} < c_{loss}$. A thorough study would be needed to fully understand the relationship between the output function norm and cross-entropy loss.
>
> **Table: Average values of different critical constants of 4 layer FCNs trained with MSE loss using SGD with batch size $512$. Each value is an average over 10 distinct initializations.**
>
> | $d$ | $w$ | $c_{loss}$ | $c_{sharp}$ | $c_{output}$ | $c_{max}$ |
> |-----|-----|------------|-------------|--------------|----------|
> | 4   | 256  | 3.05      | 3.85        | 3.04         | 7.69     |
> | 4   | 512  | 2.52      | 3.34        | 2.55         | 8.41     |
> | 4   | 1024 | 2.30      | 3.35        | 2.30         | 9.86     |
> | 4   | 2048 | 2.24      | 3.39        | 2.24         | 10.22    |
>
> ---
>
> **Table: Average values of different critical constants of 4 layer FCNs trained with cross-entropy loss using SGD with batch size $512$. Each value is an average over 10 distinct initializations.**
>
> | $d$ | $w$ | $c_{loss}$ | $c_{sharp}$ | $c_{output}$ | $c_{max}$ |
> |-----|-----|------------|-------------|--------------|----------|
> | 4 | 256   | 7.70       | 10.94       | 4.79         | 21.68    |
> | 4 | 512   | 7.03       | 13.23       | 4.59         | 37.52    |
> | 4| 1024   | 6.66       | 22.32       | 4.14         | 69.42    |
> | 4| 2048   | 5.51       | 5.83        | 3.64         | 101.80   |

---

> > ### Comment · Reviewer_bhX5 · 2023-08-10
> >
> > Thank you for your answers and additional numerical experiments.

---

### Official Review · Reviewer_BcD3 · 2023-07-06

**Soundness:** 4 excellent
**Presentation:** 2 fair
**Contribution:** 3 good
**Rating:** 7
**Confidence:** 4

**Summary:**

The paper identifies new phenomena involving the dynamics of the sharpness (largest eigenvalue of the Hessian) in the early stages of neural net training, for ReLU nets of various architectures that are initialized using He initialization.   Previously, it was known that for very wide networks, there are two 'phases' in hyperparameter space: on the one hand, if the step size exceeds 2/(initial sharpness), then a "catapult" occurs (https://arxiv.org/abs/2003.02218) and the sharpness drops; on the other hand, if the step size is below this threshold, then the sharpness remains at its initial value.  This paper fills in the picture for _narrow_ nets.   For these nets, the paper shows that the sharpness decreases at the beginning of training even when the step size is less than 2/(initial sharpness).   As a consequence, the train loss does not spike even when the step size is bigger than 2/(initial sharpness), since the sharpness drops first.   Only for larger learning rates does the train loss initially spike.  Then, for still larger learning rates, the paper shows the _sharpness_ also spikes (which the paper calls a "sharpness catapult"), which has not been observed previously.  For several architectures, the paper draws phase diagrams depicting which learning rates cause which behavior.  The paper then investigates connections to the loss connectivity literature, finding that only very large learning rates result in a barrier in the loss landscape (and that not every catapult results in a barrier).  The paper then investigates the role of initial network output, finding that initial sharpness reduction goes away if the network is made to output zero at initialization.  The paper finally reproduces all of these phenomena analytically in a simplified model (the u-v model from the catapult paper).

**Strengths:**

- The paper identifies and carefully studies a novel phenomenon involving the dynamics of the sharpness at the early stage of training.
- The experiments are very thorough (though limited to ReLU nets initialized with He initialization)
- There is an analytical explanation in a simplified model
- The findings shed some light on mode connectivity


**Weaknesses:**

- The paper doesn't explain the early-stage sharpness reduction for general neural nets (i.e. beyond the simple u-v model)

**Questions:**

- Have you looked at loss connectivity between the final parameters (at the end of training) and the initialization?

**Limitations:**

discussed above

---

> ### Author Rebuttal · Authors · 2023-08-07
>
> We thank the reviewer for their encouraging comments. Here is our response to the question on loss connectivity:
>
> > Have you looked at loss connectivity between the final parameters (at the end of training) and the initialization?
>
> As the main focus of this work is the early training dynamics, we did not analyze the loss connectivity between the initial and final parameters. However, we refer to a couple of studies that linearly interpolate the loss between the initial and final parameters.
>
> Goodfellow et al. 2014 (https://arxiv.org/abs/1412.6544) linearly interpolate the loss between the initial and final points of training and show that training does not encounter any barriers. However, they do not analyze the effect of large learning rates. A more recent study by Lucas et al. 2021 (https://arxiv.org/abs/2104.11044) analyzed the effect of large learning rates in Section 4.1 of their paper. They report that training traverses a barrier at large learning rates, which is in agreement with the naive intuition of a barrier between the initial and final points of the loss catapult. However, they do not distinguish their result as a function of different learning rate phases. We believe that analyzing such results through the lens of our phase diagram could lead to a more comprehensive understanding of these phenomena.

---

> > ### Comment · Area_Chair_kLa1 · 2023-08-18
> > **Thanks for authors' rebuttal!**
> >
> > Reviewer BcD3, did the authors address your concerns on early-stage sharpness reduction for general neural nets? Thanks.

---

> > > ### Author Response · Authors · 2023-08-21
> > > **Further clarifications on early sharpness reduction and loss catapult**
> > >
> > > Even though this question was directed at Reviewer BcD3, we would like to take the opportunity to clarify this point. We have observed two separate phenomena in general neural networks with standard parameterizations: (i) the initial reduction in sharpness (i.e. the "sharpness reduction phase") and (ii) the systematic increase of $c_{loss}$ (marking the onset of the loss catapult) with $1/w$, leading to the opening up of the sharpness reduction phase with $1/w$. To further understand these phenomena, we performed systematic experiments as discussed in Section 4 and Appendices G and H. These results indicate that the initial sharpness reduction depends on the function output scale at initialization (see Figures 29 and 31). This provides some explanation, albeit incomplete, for the origin of (i). However, the reasons behind the scaling of $c_{loss}$ with $1/w$ remain more unclear. This scaling only ceases when the function output is set to zero at initialization (Appendix G, Figure 30) and persists even when the network output is made small by scaling it by a constant (Appendix H, Figure 32).
> > > We made some progress in developing a theoretical understanding of these phenomena in a toy model (the uv model), but developing a complete theoretical understanding for general neural nets is beyond the scope of this work. Given the richness and universality of these findings across architectures and datasets, we believe it deserves the attention of the wider community to arrive at a complete understanding.

---

### Official Review · Reviewer_qjGV · 2023-07-07

**Soundness:** 2 fair
**Presentation:** 2 fair
**Contribution:** 2 fair
**Rating:** 5
**Confidence:** 3

**Summary:**

In this paper, the authors studied the effects of learning rate, width and depth of neural networks for early training dynamics. Specifically, the authors focused on the value of loss and the maximum eigenvalue of Hessian of loss and found that there are 4 different possible training dynamics in the early training. These different regimes are determined by the value of learning rate and are related with depth and width. Experiment results are provided to support the claim. A simple example with the same phenomena is also given and analyzed.

**Strengths:**

1.	Understanding the effect of large learning rate and its effect on the nonlinear training dynamics is an important problem.
2.	The paper presents many experiment results to support their findings.
3.	The finding that there exist different regimes depending on the learning rate ($c_{loss}$, $c_{sharp}$, $c_{max}$) in the early training seems to be interesting and new.


**Weaknesses:**

1.	The current paper seems to have lots of results and experiments in the main text. As a reader, it is not very easy for me to get the main conclusion for each section/set of experiments. It might be good to highlight the conclusions so that the readers can understand the point easier.
2.	The current paper tries to study the effect of depth and width on those $c$ values ($c_{loss}$, $c_{sharp}$, $c_{max}$) related with stepsize. However, it seems to me that only a few choices of width $w$ are tried in the experiments (Figure 3) and no results showing the effect of depth $d$ in the main text. It might be good to include more results on different width and depth so that the fitted curves could use more than only 3 or 4 data points and the support of the conclusion could be stronger.


**Questions:**

1.	I was wondering for these $c$ values ($c_{loss}$, $c_{sharp}$, $c_{max}$), how many times do the experiments repeat?
2.	For the definition of these $c$ values, I was wondering how time $T_1$ is chosen.
3.	In Definition 4, should $\chi_\tau$ be $\chi_\tau^\prime$?
4.	In Figure 4, I was wondering if the networks have the same depth and width, or different point represents network that may have different depth and width.


**Limitations:**

The limitations are discussed in the paper. This is a theoretical work and therefore does not seem to have negative societal impact.

---

> ### Author Rebuttal · Authors · 2023-08-07
>
> We thank the reviewer for their comments and careful reading. Below are our responses to the questions and comments:
>
> > The current paper seems to have lots of results and experiments in the main text. As a reader, it is not very easy for me to get the main conclusion for each section/set of experiments. It might be good to highlight the conclusions so that the readers can understand the point easier.
>
> We can highlight/summarize the conclusion of each experiment in each section if it helps readers to grasp the main points easily. This requires minor edits, and we can implement them in the updated version of the manuscript.
>
> > The current paper tries to study the effect of depth and width on those values ($c_{loss}, c_{sharp}, c_{max}$) related with stepsize. However, it seems to me that only a few choices of width $w$ are tried in the experiments (Figure 3) and no results showing the effect of depth $d$ in the main text. It might be good to include more results on different width and depth so that the fitted curves could use more than only 3 or 4 data points and the support of the conclusion could be stronger.
>
> For each architecture, we explored various values of depth and width, considering $10$ distinct initializations for each. Specifically, we tested $4$ values of widths and $3$ values of depths for FCNs and $3$ values of widths for CNNs. We concur that adding more values could strengthen our conclusions. Figure 1 of the PDF attached to the global response shows the phase diagrams of three neural networks with ~10 width values: (a) 8-layer FCNs trained on MNIST trained with MSE, (b) 7-layer CNNs trained on Fashion-MNIST using MSE, and (c) 16-layer FCNs trained on CIFAR-10 using MSE. We will make these adjustments in the revised version. We also agree that including the effect of depth in the main text would be beneficial and will make these enhancements in the updated version of the manuscript.
>
> > I was wondering for these $c$ values ($c_{loss}, c_{sharp}, c_{max}$), how many times do the experiments repeat?
>
> As mentioned in line 4 of Figure 3 caption, each experiment is repeated for $10$ distinct initializations for each depth and width to obtain the $c$ values.
>
> > For the definition of these $c$ values, I was wondering how time $T_1$is chosen.
>
> We choose $T_1$ to be the smallest value of step $t$ that contains the entire duration of the catapult effect. For the widths and depths considered, the catapult typically lasts at most $\sim 10$ steps. Thus, for computational efficiency, we've set $T_1 = 10$.
>
> > In Definition 4, should $\chi_\tau$ be $\chi_\tau'$?
>
> We thank the reviewer for pointing this out. We have fixed this in the updated version of the manuscript.
>
> > In Figure 4, I was wondering if the networks have the same depth and width, or different point represents network that may have different depth and width.
>
>  In Figure 4, each data point can correspond to a run with varying depth, width, and initialization. With $n_w$ values of width, $n_d$ values of depth, and $10$ initializations, there are a total of $n_w \times n_d \times 10$ data points in this Figure, which illustrates that the inequality $c_{loss} \leq c_{sharp} \leq c_{max}$ holds regardless of depth, width, and initialization.  We thank the reviewer for pointing this out. We will update the Figure caption to include this information in the revised version of the manuscript.

---

> > ### Comment · Reviewer_qjGV · 2023-08-13
> >
> > Thanks for the response to address my question. It makes things clear and help me have a better understanding of the paper. I will increase my score.

---

### Official Review · Reviewer_bXcV · 2023-07-08

**Soundness:** 3 good
**Presentation:** 4 excellent
**Contribution:** 3 good
**Rating:** 6
**Confidence:** 3

**Summary:**

This work mainly investigates the learning dynamics depending on the learning rate and categorizes its qualitative behaviors in some phases. In particular, the authors identify new phases; sharpness reduction, and loss-sharpness catapult. They investigate the dependence of the critical learning rates (mainly, $c_{class}, c_{sharp}, c_{max}$ and $c_{barrier}$) on the width, depth & loss type, and try to identify a universal relationship between them.
Their empirical observations in some deep neural networks are also confirmed in a simple linear network and the obtained phase diagram is expected to hold in various situations.

**Strengths:**

This work attacks the challenging problem to find any universal behavior of early learning dynamics in finite-size networks (I guess that the terminology of "phase diagram" comes from physics or chemistry but such a dynamic and finite-sized diagram seems not known). Despite this difficulty, the authors verified new phases (characterized by $c_{loss},c_{sharp},c_{max}$) in various settings and these new phases seem to have the originality. Their careful and elaborated experiments are persuasive and enhance the quality of the paper.


**Weaknesses:**

(a) The whole findings are empirical. Even in a simple model of $uvx$, the authors did not solve the dynamics in an analytical form. It is quite hard to obtain any intuition on why the three phases characterized by $c_{loss}<c_{sharp}<c_{max}$ universally appear.

 (b)They are several unclear points that the authors should more clearly mention. See below.


**Questions:**

**Connection to the infinite width limit**

While the current work focuses on finite-sized networks, some previous work investigated the infinite-width limit [64]. In particular, [64] claims that for the infinite-width limit, the learning dynamics *must be* the kernel regime for SP initialization (nevertheless, the learning dynamics does not progress or explodes).  How is this infinite-width study related to your results? Since the critical learning rate of the kernel regime is given by $2/\lambda_{0}^{NTK}$, does this suggest that all  $c_{loss,sharp,max}$ converge to 2 in the infinite width limit?

**Phase diagram in previous work**

Related to [64], there are several studies on the phase diagram of convergence in the infinite width case (e.g. https://arxiv.org/abs/2012.15110 and https://arxiv.org/abs/2007.07497). It would be better to mention them for giving a rich overview of the phase-diagram studies in the literature.  More related to the current work, the phase diagram for width v.s. learning rate is also investigated in the literature (https://arxiv.org/abs/1806.01316). This work implies that 2/$\lambda_{0}^{NTK}$ works as a critical learning rate of the convergence in the infinite width.

**The effect of early learning dynamics on eventual minima**

I understand that the main focus of the current work is on the early training regime, but it is quite curious how the sharpness reduction, loss catapult, and loss-sharpness catapult phases determine the eventual generalization performance and sharpness of minima in the trained models. If this point would be clarified, it would increase the significance of this paper much more.

**Limitations:**

**No normalization layers**

The architectures in this work have no batch/layer normalization layer. Since the batch norm is known to strongly affect the sharpness (e.g. https://arxiv.org/abs/1901.10159), it is curious how the normalization layer potentially changes the results.

**Difference between MSE and cross-entropy cases**

It seems that the cross-entropy loss shows a different behavior of the critical learning rates. In particular, in most cases, $c_{loss}$ and $c_{sharp}$ decrease close to 2 as $1/w$ decreases in the MSE case while they keep far from 2 in the cross-ent case. In the revised version, I believe it would be better to give a more elaborated explanation of the difference in the main text.

---

> ### Author Rebuttal · Authors · 2023-08-07
>
> We thank the reviewer for taking the time to read our manuscript and for providing detailed comments. Below are our responses to the comments provided:
>
> > Connection to the infinite width limit:
>
> In our study, we are looking at the behavior of the optimization dynamics at training timescales $t_\star$ that grow with width $w$. Specifically, the end of the early time transient period occurs at $t_\star \sim \log(w)$. However, in the usual infinite width studies in SP initialization [64], the training dynamics is restricted to $O(1)$ times in the limit of infinite width; this leads to the kernel regime, for which the system is in a lazy training regime for learning rates less than $2/\lambda_0^{NTK}$ and divergent training for larger learning rates. Our results for critical constants in the limit of large width, therefore, do not converge to $2$ as would be expected from the analysis of [64] (except for $c_{loss}$, which converges to $2$ for MSE loss). We will provide a more detailed explanation in the updated version of the paper to clarify this distinction.
>
> > Phase diagram in previous work:
>
> We will incorporate the above suggestion by including a paragraph discussing the phase diagram from previous studies in the updated version of our manuscript.
>
> > The effect of early learning dynamics on eventual minima:
>
> We agree that analyzing generalization properties through the lens of the phase diagram could lead to a more comprehensive understanding and enhance the importance of the study. However, this is a non-trivial task due to the extensive hyperparameter search space (including depth, width, learning rates, initializations, and batch size) and the significantly large training time required for each workload (model and training task). Moreover, defining the success measure, such as finding the best generalization error for a fixed compute versus the shortest training time for a given target error, adds another layer of complexity. A comprehensive analysis would result from a cumulation of studies, with our study contributing to this effort.
>
> > Difference between MSE and cross-entropy cases:
>
> The disparity in the limiting value of $c_{loss}$ at large width is mostly due to the non-constant Hessian of the cross-entropy loss, as discussed in [62]. Previous work, such as [47], analyzed the catapult dynamics for the $uv$ model with logistic loss and demonstrated that the loss catapult occurs above $\eta_{loss} = 4 / \lambda_0^{NTK}$.
>  Their argument is as follows.
>
>  Consider the $uv$ model trained on a binary classification task using the logistic loss on two training examples $ (x_1, y_1) = (1, 1) $ and $ (x_2, y_2) = (1, -1) $. Then, the total loss is $ \mathcal{L}(f) = \frac{1}{2} \log (2 + 2 \cosh(f)) $. Hence, the loss grows monotonically as the output function $ |f| $ increases.
> The update equation of the function is given by:
>
> \begin{align}
>     f_{t+1} = f_t \left( 1 -  \frac{\eta \lambda^{NTK}_t \mathcal{L}'(f)}{f_t} + \frac{\eta^2 \mathcal{L}'(f_t)^2}{w}  \right),
> \end{align}
> where $\eta$ is the learning rate, $w$ is the width, and $\mathcal{L}'(.)$ is the derivative of the loss.
> At large width $w$, if the condition $ | 1 - \eta \lambda^{NTK} \mathcal{L}'(f_t) / f| < 1 $ holds, then the output function continues to decrease. Given that $ \mathcal{L}'(f) / f \leq 1/2 $ in the above case, this decrease persists for $ \eta \lambda^{NTK} < 4 $.
>
> This result provides some intuition behind the discrepancy. We can describe the above analysis in more detail in the updated version of the manuscript. However, a complete understanding of the catapult phenomenon in the context of cross-entropy loss requires a more detailed examination.

---

> > ### Comment · Reviewer_bXcV · 2023-08-14
> >
> > Thank you for your kind reply. I am looking forward to seeing the accepted version, which will briefly include the above answers.
> >
> > >a complete understanding of the catapult phenomenon in the context of cross-entropy loss requires a more detailed examination.
> >
> > I agree. I hope that this point will be clarified in your next work or other subsequent works.

---

### Author Rebuttal · Authors · 2023-08-08

We would like to express our gratitude to all the reviewers for taking the time to review our paper and providing detailed comments. Attached below is the PDF containing the figures required for some responses.

---

### Decision · Program_Chairs · 2023-09-21

**Decision:**

Accept (poster)

**Comment:**

The paper studies empirically nonlinear training dynamics under different learning rate, network width and height. It is an important topic and reviewers agree that the paper is well-written and the experiments are relatively thorough. Therefore I vote for acceptance.